# Evaluating the long-term strength of GGBFS-blended cement across various water-to-binder and superplasticizer ratios under heating/cooling cycles

Bashdar Omer[1], Najmadeen Mohammed Saeed[1], Mahmood Hunar Dheyaaldin[2,3], Ahmed Salah Jamal[4], Rawaz Kurda[5]*

**1** Civil Engineering Department, University of Raparin, Ranya, Iraq, **2** Research Associate of the Department of Civil Engineering, American University in Dubai, Dubai, PO Box 28282, UAE, **3** Civil Engineering Department, Faculty of Engineering, Tishk International University, Erbil, Iraq, **4** Department of Civil Engineering, American University in Dubai, Dubai, UAE, **5** Department of Highway and Bridge Engineering, Technical Engineering College, Erbil Polytechnic University, Erbil, Iraq

* Rawaz.kurda@epu.edu.iq

## Abstract

The investigation is to understand the combined effects of ground granulated blast furnace slag (GGBFS) content, superplasticizer (SP) dosages, and water-to-binder (w/b) ratios on the fresh and harden properties, as well as long-term durability under thermal cycles. This study aims to address these gaps by examining the influence of varying GGBFS replacement levels, SP dosage, and w/b ratios on blended mortar's fresh properties, mechanical strength, thermal performance, water absorption, and microstructure. The experimental method included testing for flowability, fresh density, and compressive strength over 56 and 90 days and 30% for 365 days and evaluating the microstructure through microscopy. The result showed that increasing GGBFS content improved both flowability and fresh density. Additionally, thermal cycling led to a significant increase in compressive strength with an average strength gain of 22% at 56 days, 40.9% at 90 days, and 30% at 365 days. The microstructure, which demonstrates a lot of calcium silicate hydrate (C-S-H) crystals and is relatively denser after 365 days, results in high strength. In conclusion, incorporating GGBFS in cement composites reduces $CO_2$ emissions, improves performance, and enhances durability, especially under thermal conditions, making it a viable solution for providing high-performance concrete. This finding has significant implications for reducing the environmental footprint and life cycle in construction.

## 1. Introduction

Ordinary Portland Cement (OPC), which has served as the fundamental foundational material driving the progress of modern human civilization, has a nearly (200)-year history. However, the manufacture of OPC involves considerable $CO_2$ emissions, with the cement sector accounting for 11% of all carbon emissions worldwide [1,2]. Physicochemical characteristics of their parent material GGBFS. Modernization and globalization are increasingly producing enormous amounts of

**Data availability statement:** All relevant data are within the manuscript.

**Funding:** The author(s) received no specific funding for this work.

waste, and properly disposing of such waste poses major risks. Researchers have been looking into using alternative binders to address cement's sustainability problems and decrease the environmental impact of construction industries. Utilizing waste as supplementary cementitious materials is essential in mitigating the perils of global warming and industrial waste accumulation. It also aids in reducing cement consumption and curbing the detrimental environmental impacts associated with cement production [3,4]. Moreover, cement is the most expensive and energy-intensive among all concrete components. Concrete manufacturing costs are decreased by using industrial waste that contains important pozzolanic minerals as partial replacements. This involves incorporating mineral admixtures like silica fume, fly ash, GGBFS, and others. This pozzolan material into concrete not only reduces the quantity of cement necessary but improves concrete characteristics as well [5–7]. According to Alamri et al [8], the incorporation of supplementary cementitious materials with OPC has faced growing regulatory scrutiny in the construction industry, as evidenced by multiple international standards [9]. Therefore, pozzolanic additives can be used to improve the durability properties of concrete and mortar.

Due to its capacity to be shaped at room temperature and other outstanding attributes, cement has experienced significant success within the construction industry [10]. The use of ordinary Portland cement (OPC) significantly impacts the environment due to the high energy consumption and greenhouse emissions during its production [11]. One ton of cement production emits about 1 ton of $CO_2$. Cement demand will likely increase even further as the building sector expands dramatically. By 2050, global cement demand is expected to increase by almost 400% compared to the 1990s [12]. One and a half tons of virgin materials are required to produce one ton of cement, which additionally affects the environment but also consumes a lot of virgin resources [13].

One of the acknowledged superior alternatives to cement in a number of processes is GGBFS. It is a by-product material of iron that is removed from blast furnaces and processed into a very fine powder using a revolving ball mill [14,15]. The significance of GGBFS lies in its role as an eco-friendly substitute for conventional concrete components. Its use contributes to energy conservation, reduces $CO_2$ emissions, and helps in the preservation of natural resources when employed as a construction material. It has been demonstrated that BFS's roughness and dispersive properties improve workability and reduce concrete's permeability capacity and density [16]. However, the optimal use of GGBFS in mortar requires a proper understanding of the impact of its replacement rate on the mortar properties. While a substantial addition of GGBFS may decrease the compressive strength of concrete, it enhances various other attributes such as permeability, water absorption, resistance to acid penetration, and more [16]. Typical (GGBFS) replacement rates in GGBFS blended cement can range from 25% up to 85%, however, 50% is typically employed in most applications [17–20]. Kumar et al. [21] studied the durability characteristics of blended cement made by substituting slag for OPC at a weight ratio of 30%, 50%, and 70%. Their research revealed that GGBFS and alternative materials augmented the mechanical strength and durability properties of cement mortar. The specimens with GGBFS presented the dense microstructural under SEM analysis with combined good bonds between aggregate and binders [22]. Replacing cement with GGBFS decreases the amount of cement, which reduces heat release, shows the hydration process, and lowers the intensity of heat generation. The dilution effect in the mix speeds up the reaction as the GGBFS substitution rate increases from 50% to 65% [23]. The average compressive strength and tensile strength of mixes that contain GGBFS replacement were similar to those of OPC. Mixes with 50% replacement and higher GGBFS contents demonstrated greater compressive strength compared to those with 70% replacement [24]. Calcium formate has proven to be an effective accelerator, significantly improving the properties of mechanical and microstructural of CaO activated ultra-high performance concrete-based GGBFS [25,26].

On the other hand, the w/b ratio and SP content significantly affect the properties of (GGBFS) blended cement mortars. A high w/b ratio leads to decreased strength and heightened porosity in the mortar, whereas a low w/b ratio can yield higher strength but increased brittleness. Strength and durability can be increased by incorporating an SP to the mix, which can assist keep workability while lowering the w/b ratio. When SP is present in high concentrations, however, it can make the mortar excessively fluid, which can result in increased bleeding, segregation, and compromised mortar. Moreover, the mortar's setting time and ultimate strength can be determined by the kind and amount of SP used. Consequently, meticulous control over the w/b ratio and SP content is essential to attain optimal properties in (GGBFS) blended cement mortars. In modern concrete production, SPs play a vital role as they improve the mechanical strength and durability of concrete by reducing the required water content during preparation. The absorption of SP onto the cement particles produces a delay in the setting time [27]. However, for the interaction of SP with (GGBFS), it has been reported that the amount of C3A is reduced as a result of replacing OPC with GGBFS leaving more SP to disperse the cement particles [28]. Yakobu et al. [29] conducted a research study to explore the compatibility of cement mixtures with varying GGBFS contents, employing two distinct types of SP to determine the ideal dosage through the marsh cone test. Their findings indicated that the SP quantity could be decreased relative to the increase in GGBFS inclusion. Additionally, they observed that using polycarboxylate ether-based SP yielded superior strength outcomes compared to sulphonated naphthalene formaldehyde-based SP. Additionally, according to Al-Tayyib et al., the compressive strength and flexural strength can be reduced by 32% and 27%, respectively, during heating-cooling cycles, this can significantly affect durability [30]. Saeed et al. [31] demonstrated that the compressive strength of cement mortars is significantly influenced by the w/b ratio and GGBFS content. The study found that incorporating GGBFS improved the compressive strength, with the optimal performance observed at a w/b ratio of 0.44 and 22.5% GGBFS replacement. Specifically, the highest compressive strength of 47.4 MPa was achieved with 22.5% GGBFS, 2% SP, and at an elevated temperature of 225°C. The mechanical behavior of the cement mortar was notably enhanced by the interaction between w/b ratios and supplementary cementitious materials like GGBFS.

These environmentally friendly additives are used to minimize construction costs, partially replace cement, and make concrete an eco-friendly material [32]. As a result, research is required to identify more environmentally friendly materials that can deliver performance that is either comparable to or superior to that of traditional materials in the same or similar settings. Thus, the combination of industrial or agricultural wastes with cement, either completely or partially, has been encouraged to reduce the use of cement. One of these materials, known as GGBFS, has proven to be effective for such procedures [33]. The comparison of the two values, 913 for OPC and 67 for GGBFS demonstrates that SCMs like GGBFS have much lower $CO_2$ emissions than OPC, making them excellent choices for sustainable concrete [31]. However, there are a number of sustainability issues connected to the use of GGBFS: [34] GGBFS is dependent on the amount of production from the steel and iron industries because it is a byproduct of these industries. The use of GGBFS in the manufacturing of concrete may be impacted by the restricted availability of GGBFS during periods of low production. In particular, if GGBFS is carried across large distances, it can be a significant source of greenhouse gas emissions. GGBFS from regional or local sources can help to reduce this. The composition of GGBFS, can change depending on the production process. As a result, the performance of concrete containing GGBFS may be affected. This may influence the quality and uniformity of GGBFS. In some nations, the usage of GGBFS may be hindered by the fact that it is frequent.

Today, (GGBFS) is extensively used in the construction of normal-strength and high-performance concrete [35]. Furthermore, the significance of the mentioned parameters, and

comprehending their collective impact on blended cement is crucial. However, after conducting an in-depth literature review, it was revealed that there is little to no data on the evaluation of GGBFS blended cement under different (GGBFS) replacement contents, w/b ratios, and SP dosages. Therefore, this research studies the sustainable challenges of cement mortar substituted by (GGBFS) with different w/b ratios and SP contents.

The research highlights several gaps for further investigation. Previous studies have examined the substitution of GGBFS for cement at varying percentages. However, this research uniquely explores more diverse and extreme thermal cycling conditions to better simulate real-world scenarios in different climate regions. Future studies should address the impact of freeze-thaw cycles on the durability and performance of GGBFS-based cement composites, as these are critical durability factors. A detailed life cycle assessment is necessary to quantify the reduction of $CO_2$ emissions.

The main objective of this work is to thoroughly investigate how different w/b ratios affect the compressive performance and long-term strength of cement modified with (GGBFS) under a variety of heating and cooling conditions. 30 cyclic heating and cooling processes will be used in the study to test various temperatures in order to incorporate ongoing challenges present in the real world [36–38].

This research aims to:

I. Investigate the effects of various w/b ratios on the compressive strength of cement modified with GGBFS that have experienced cyclic heating and cooling processing at various temperatures.

II. Correspond the compressive strength statistics with the specified w/b ratios to evaluate the long-term strength development of the modified cement at certain age intervals.

III. Perform statistical analysis on ANOVA and comparison between actual and prediction results.

## 2. Experimental program

The performance of GGBFS blended cement with various mixture design parameters was investigated experimentally in this study. Fig 1 depicts a schematic of the experimental process. In this section, Materials (section 2.1:materilas), mix design (section 2.2: mix design), the preparation method of the samples (section 2.3: preparation method of the samples), testing methods (section 2.4: testing methods), cyclic heating, and cooling of compressive strength (section 2.5: cyclic heating, and cooling of compressive strength) have been described in detail.

### 2.1. Materials

The binders employed in this study included Ordinary Portland Cement (OPC) from Tasulja cement factory in Iraq with a specific gravity of 3.10 and GGBFS [39,40] locally manufactured by DCP construction material company with a specific gravity of 2.90, the bulk density is 1000-1100 kg/m³. Table 1 details the chemical compositions of both OPC and GGBFS. Normal river sand with a grain size distribution was utilized as the fine aggregate demonstrated in Fig 2. The SP is locally available from the DCP construction material company utilized in this research is polycarboxylate-based.

### 2.2. Mix design

In this study, as shown in Table 2, a total of 27 mixtures were created, employing different GGBFS replacement levels (0%, 22.5%, and 45%), w/b ratios (0.4, 0.44, and 0.48), and SP

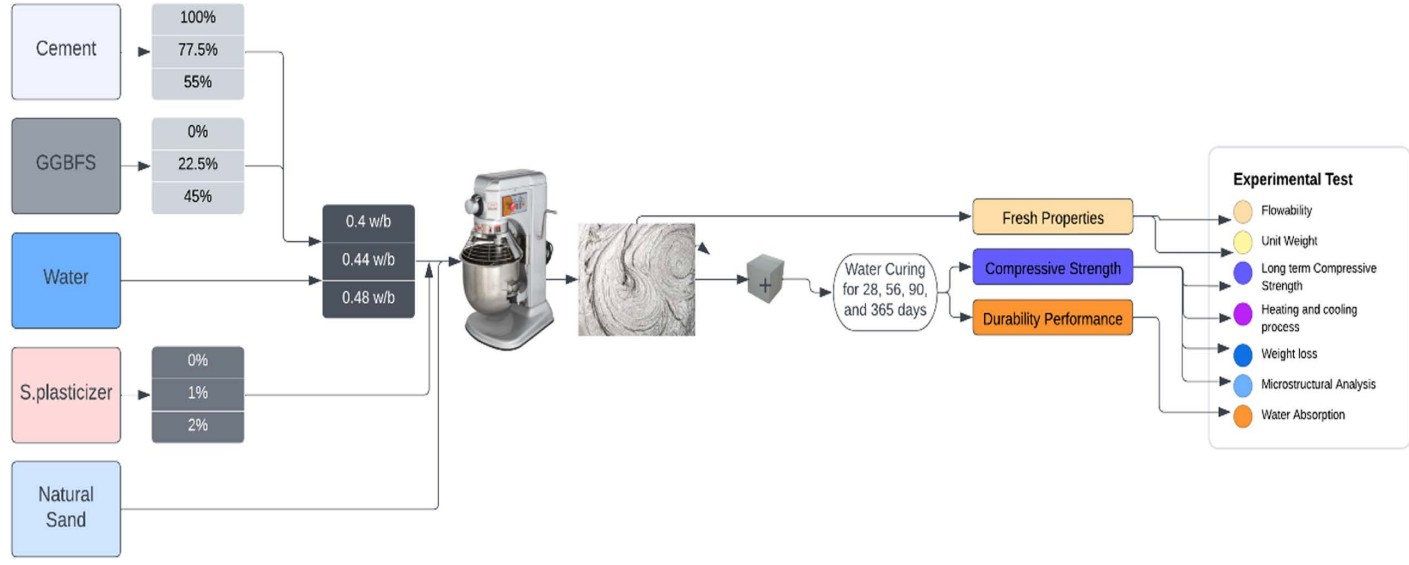

**Fig 1. A schematic of the experimental process.**

**Table 1. Chemical composition of OPC and GGBFS.**

| Component | $Al_2O_3$ | $SiO_2$ | $Fe_2O_3$ | CaO | $SO_3$ | MgO | $TiO_2$ | $K_2O$ | $Na_2O$ | $P_2O_5$ | MnO | Lol |
|---|---|---|---|---|---|---|---|---|---|---|---|---|
| OPC (%) | 5.27 | 21.27 | 3.20 | 62.40 | 2.01 | 1.61 | 0.26 | 0.52 | 0.24 | 0.05 | 0.08 | 2.55 |
| GGBFS (%) | 8.05 | 33.97 | 2.36 | 38.88 | 1.01 | 3.43 | 2.07 | 0.58 | 0.06 | 0.06 | 2.36 | 0.7 |

percentages (0%, 1%, and 2%). The overall binder content (comprising OPC and GGBFS) was adjusted and categorized into three groups, as depicted in Table 2.

### 2.3. Preparation method of the samples

The mortar mix was produced following ASTM C305 [41] guidelines, after which it was poured into cube molds to form fresh mortar specimens. These specimens were then removed after 24 hours and subjected to water curing for 28, 56, 90, and 365 days, as depicted in Fig 3.

### 2.4. Testing methods

The testing method conducted three properties that are divided into sub-tests. All following tests have been evaluated under international standards. The flowability and unit weight of the fresh mortar were measured in accordance with ASTM C1437 [42] and ASTM C138 [43], respectively. The compressive strength was determined according to ASTM C109 [44]. Using Cube molds with dimensions of 50 mm, the water absorption of mortar specimens at 28 and 56 days was determined in line with ASTM C1403 [45]. At each age, three samples were examined for water absorption, and the overall water absorption rate was determined by taking the mean value. The importance of designing mixtures that account for material behavior under temperature variation to ensure mechanical and durability performance [38].

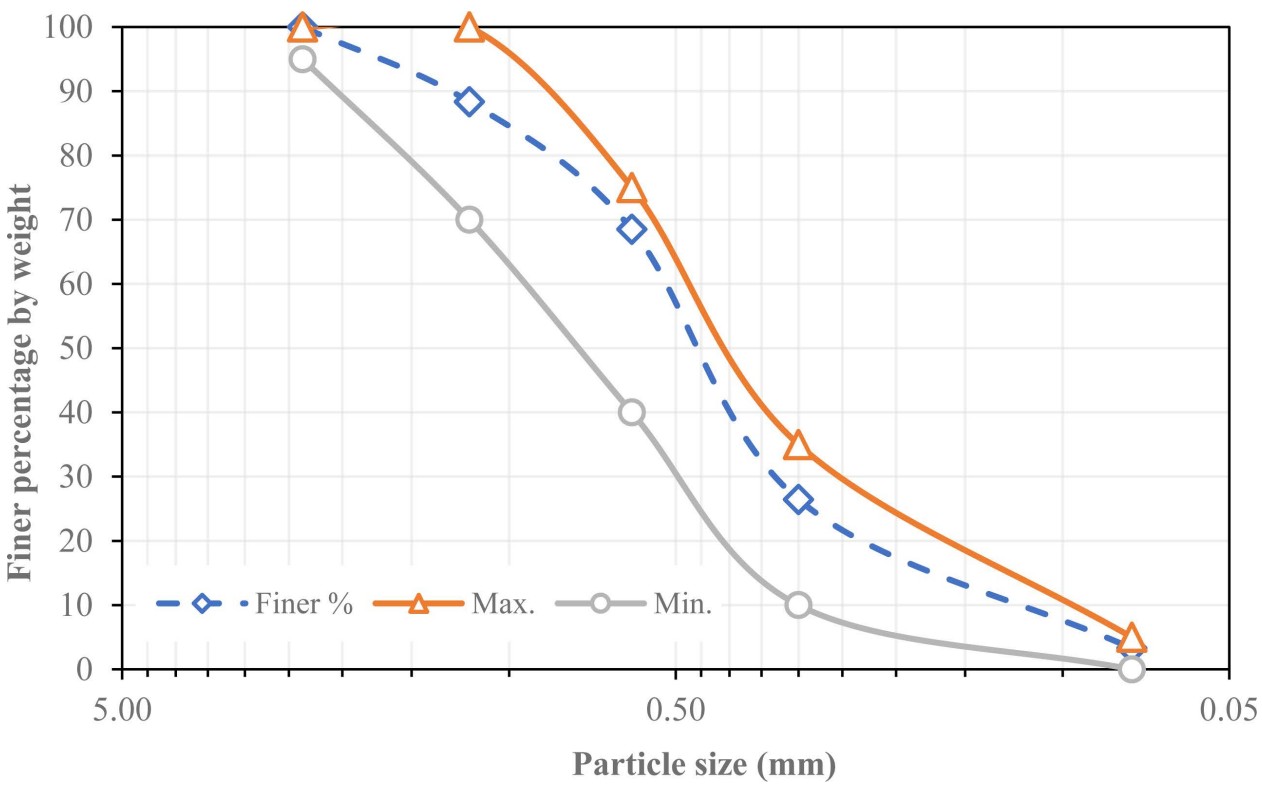

**Fig 2. Particle size distribution of natural sand.**

### 2.5. Cyclic heating and cooling of compressive strength

The specimens were subjected to heating temperatures of 50°C, 100°C, and 150°C, with each sample undergoing 30 cycles of heating and air cooling. During the heating process, In a furnace, the sample was gradually heated to the desired temperature at a rate of 8 °C per minute [30,35]. The furnace temperature was then maintained constantly at the target temperature for 2 hours. The specimens were then subjected to air cool at 25 °C after being gradually cooled. The temperature was maintained at each temperature for 1 hour to reach a thermally stable state. The oven door was then opened and the samples were allowed to cool naturally to room temperature [35]. The importance of designing mixtures that account for material behavior under temperature variation to ensure durability performance [38].

## 3. Result and discussion

### 3.1. Fresh properties

**Flowability.** The w/b and SP dosage's effects on cementitious mortar's workability are depicted in Fig 4. The flowability of cementitious mortar at different w/b ratios showed a significant pattern. Significantly, when compared to other compositions, the mortar composition with a 0.4 w/b ratio had the lowest flowability of all the ratios evaluated. Additionally, a noteworthy finding emerged when examining the impact of w/b ratios in combination with the absence of an SP. It has been shown that adding an SP significantly improves mortar's flowability. More particular, flowability was significantly improved by the addition of 1% SP to the mixture, which successfully saturated the w/b ratio. With the aid of

**Table 2. Mix proportions and the required mortar quantity by weight.**

| Groups | No | Mixes | Binder | OPC (g) | GGBFS (g) | Natural Sand (g) | SP (g) | Water (g) |
|---|---|---|---|---|---|---|---|---|
| G1 | M1 | M-C100-w/b 0.4-SP 0 | Cement 100 (%) | 2204.0 | 0.0 | 6080.0 | 0.0 | 881.6 |
| | M2 | M-C100- w/b 0.44- SP 0 | | 2204.0 | 0.0 | 6080.0 | 0.0 | 969.8 |
| | M3 | M-C100- w/b 0.48- SP 0 | | 2204.0 | 0.0 | 6080.0 | 0.0 | 1057.9 |
| | M4 | M-C100- w/b 0.4- SP 1 | | 2204.0 | 0.0 | 6080.0 | 22.0 | 881.6 |
| | M5 | M-C100- w/b 0.44- SP 1 | | 2204.0 | 0.0 | 6080.0 | 22.0 | 969.8 |
| | M6 | M-C100- w/b 0.48- SP 1 | | 2204.0 | 0.0 | 6080.0 | 22.0 | 1057.9 |
| | M7 | M-C100- w/b 0.4- SP 2 | | 2204.0 | 0.0 | 6080.0 | 44.1 | 881.6 |
| | M8 | M-C100- w/b 0.44- SP 2 | | 2204.0 | 0.0 | 6080.0 | 44.1 | 969.8 |
| | M9 | M-C100- w/b 0.48- SP 2 | | 2204.0 | 0.0 | 6080.0 | 44.1 | 1057.9 |
| G2 | M1 | M-C77.5-G22.5- w/b 0.4- SP 0 | Cement 77.5(%)-GGBFS 22.5(%) | 1708.1 | 495.9 | 6080.0 | 0.0 | 881.6 |
| | M2 | M-C77.5-G22.5- w/b 0.44- SP 0 | | 1708.1 | 495.9 | 6080.0 | 0.0 | 969.8 |
| | M3 | M-C77.5-G22.5- w/b 0.48- SP 0 | | 1708.1 | 495.9 | 6080.0 | 0.0 | 1057.9 |
| | M4 | M-C77.5-G22.5- w/b 0.4- SP 1 | | 1708.1 | 495.9 | 6080.0 | 22.0 | 881.6 |
| | M5 | M-C77.5-G22.5- w/b 0.44- SP 1 | | 1708.1 | 495.9 | 6080.0 | 22.0 | 969.8 |
| | M6 | M-C77.5-G22.5- w/b 0.48- SP 1 | | 1708.1 | 495.9 | 6080.0 | 22.0 | 1057.9 |
| | M7 | M-C77.5-G22.5- w/b 0.4- SP 2 | | 1708.1 | 495.9 | 6080.0 | 44.1 | 881.6 |
| | M8 | M-C77.5-G22.5- w/b 0.44- SP 2 | | 1708.1 | 495.9 | 6080.0 | 44.1 | 969.8 |
| | M9 | M-C77.5-G22.5- w/b 0.48- SP 2 | | 1708.1 | 495.9 | 6080.0 | 44.1 | 1057.9 |
| G3 | M1 | M-C55-G45- w/b 0.4- SP 0 | Cement 55(%)-GGBFS 45(%) | 1212.2 | 991.8 | 6080.0 | 0.0 | 881.6 |
| | M2 | M-C55-G45- w/b 0.44- SP 0 | | 1212.2 | 991.8 | 6080.0 | 0.0 | 969.8 |
| | M3 | M-C55-G45- w/b 0.48- SP 0 | | 1212.2 | 991.8 | 6080.0 | 0.0 | 1057.9 |
| | M4 | M-C55-G45- w/b 0.4- SP 1 | | 1212.2 | 991.8 | 6080.0 | 22.0 | 881.6 |
| | M5 | M-C55-G45- w/b 0.44- SP 1 | | 1212.2 | 991.8 | 6080.0 | 22.0 | 969.8 |
| | M6 | M-C55-G45- w/b 0.48- SP 1 | | 1212.2 | 991.8 | 6080.0 | 22.0 | 1057.9 |
| | M7 | M-C55-G45- w/b 0.4- SP 2 | | 1212.2 | 991.8 | 6080.0 | 44.1 | 881.6 |
| | M8 | M-C55-G45- w/b 0.44- SP 2 | | 1212.2 | 991.8 | 6080.0 | 44.1 | 969.8 |
| | M9 | M-C55-G45- w/b 0.48- SP 2 | | 1212.2 | 991.8 | 6080.0 | 44.1 | 1057.9 |

the 2% SP inclusion, the flow characteristics were successfully improved and an appropriate composing balance was achieved, promoting consistent flowability and the binding qualities required for long-lasting and reliable construction materials. Then for the 2% SP, the flowability increases, and even for 0.4 w/b ratios, the flowability is 77%, which is moderate with full cement.

The addition of SP was found to have a significant effect on the flowability, which was increased with increasing SP. The maximum flowability was obtained with 0.48 w/b, 2% SP, and 45% GGBFS. It can be shown that higher GGBFS replacement content led to higher flowability in all mixes. Alexander et al, 2022, reported that 2 percent of SPs show significant performance due to the dispersion of particles and the relative fluidity improved with an increase in SPs [46]. Das et al 2023, demonstrated that partial replacement of cement by (GGBFS) improved the performance of fluidity of concrete [47]. In addition, a trend is observed that suggested that increased GGBFS replacement content was positively correlated with increased flowability in all mortar compositions. This highlights the critical feature of GGBFS in improving flowability combined with SP concentrations in the mortar matrix. It should be mentioned that according to the analysis of the figures, higher content of SP in GGBFS-contained mixes resulted in a higher flow rate compared to OPC mixes. According to Palacios

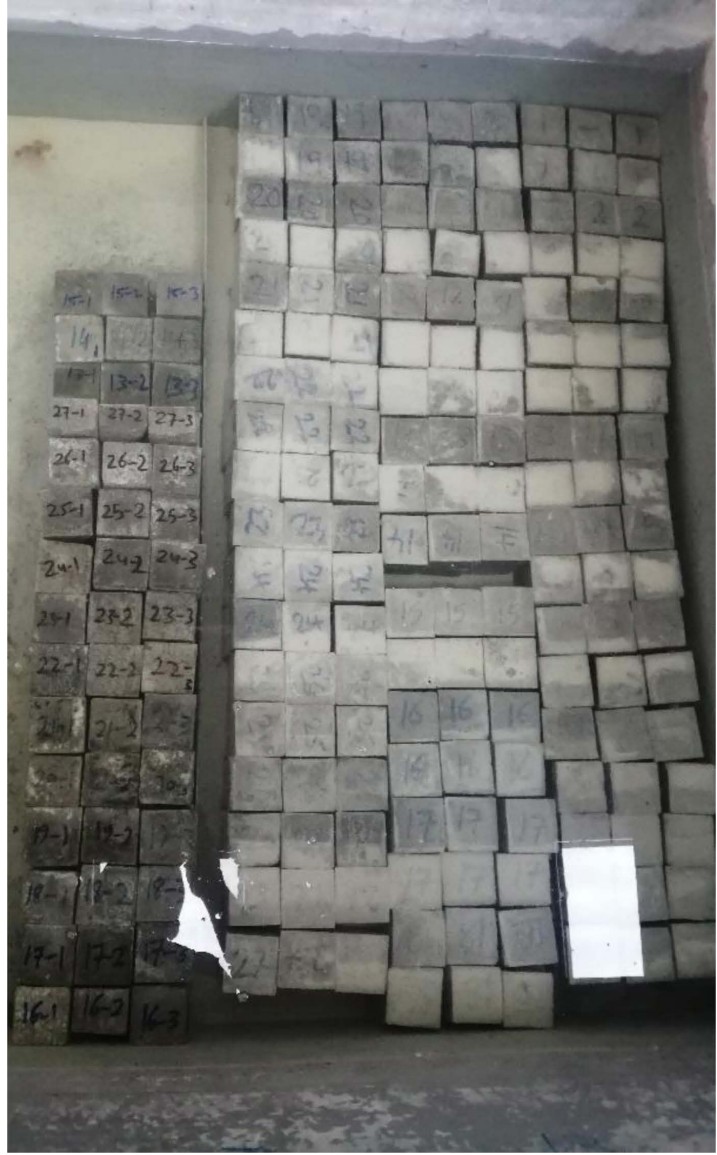

**Fig 3. Curing of samples until 365 days at 23±3°C.**

et al. [48], this is explained by the improved fluidizing characteristics of polycarboxylate-based admixtures in GGBFS blended cement as opposed to actual cement.

**Fresh unit weight.** A comprehensive examination of the fresh unit weight variations in GGBFS-modified cement mortar over different GGBFS percentages, w/b ratios, and SP levels of 0%, 1%, and 2% are shown in Figs 5–7. Fig 5 shows that how different mixes of (OPC) and (GGBFS) affected the unit weight of concrete at varying water-to-binder (w/b) ratios (0.4, 0.44, 0.48) without SP. As GGBFS replaces OPC, the unit weight tends to decrease, particularly at higher w/b ratios. However, the unit weight increases with higher water content in all mixes, though GGBFS has a more noticeable influence in reducing density at higher replacement levels. This suggests that GGBFS can be used to reduce cement content without significantly compromising unit weight, it is an eco-friendly option for concrete production.

Fig 6 demonstrates how 1% of SP affects the unit weight of concrete mixes with varying proportions of OPC and GGBFS at different w/b ratios. With SP, unit weight generally increases across all mixes, especially at higher w/b ratios. The use of GGBFS still reduces unit weight, but the SP helps maintain higher density compared to the 0% SP. This suggested that adding SP improves workability and compaction, leading to denser mixes, even when GGBFS is partially replaced.

Furthermore, with 2% of SP, the unit weight of all mixes is generally higher showed in Fig 7. The 0.4 w/b mix shows the most significant gain in density, especially with the partial GGBFS replacement. The increase in unit weight reflects the improved workability and compaction provided by SP, which counteract the potential reduction in density from GGBFS substitution. However, at higher w/b ratios, particularly with larger GGBFS content, the unit weight declines slightly.

## 3.2. Bulk density

Various GGBFS replacement levels, w/b ratios, and SP doses were included in the bulk density evaluation for blended cement are depicted in Figs 8–10. A comprehensive examination of these figures demonstrates a consistent trend: an escalation in GGBFS replacement content consistently diminishes the bulk density across all mix designs and w/b ratios, regardless of the variations in SP dosages. The results indicate that the bulk density of the blended cement

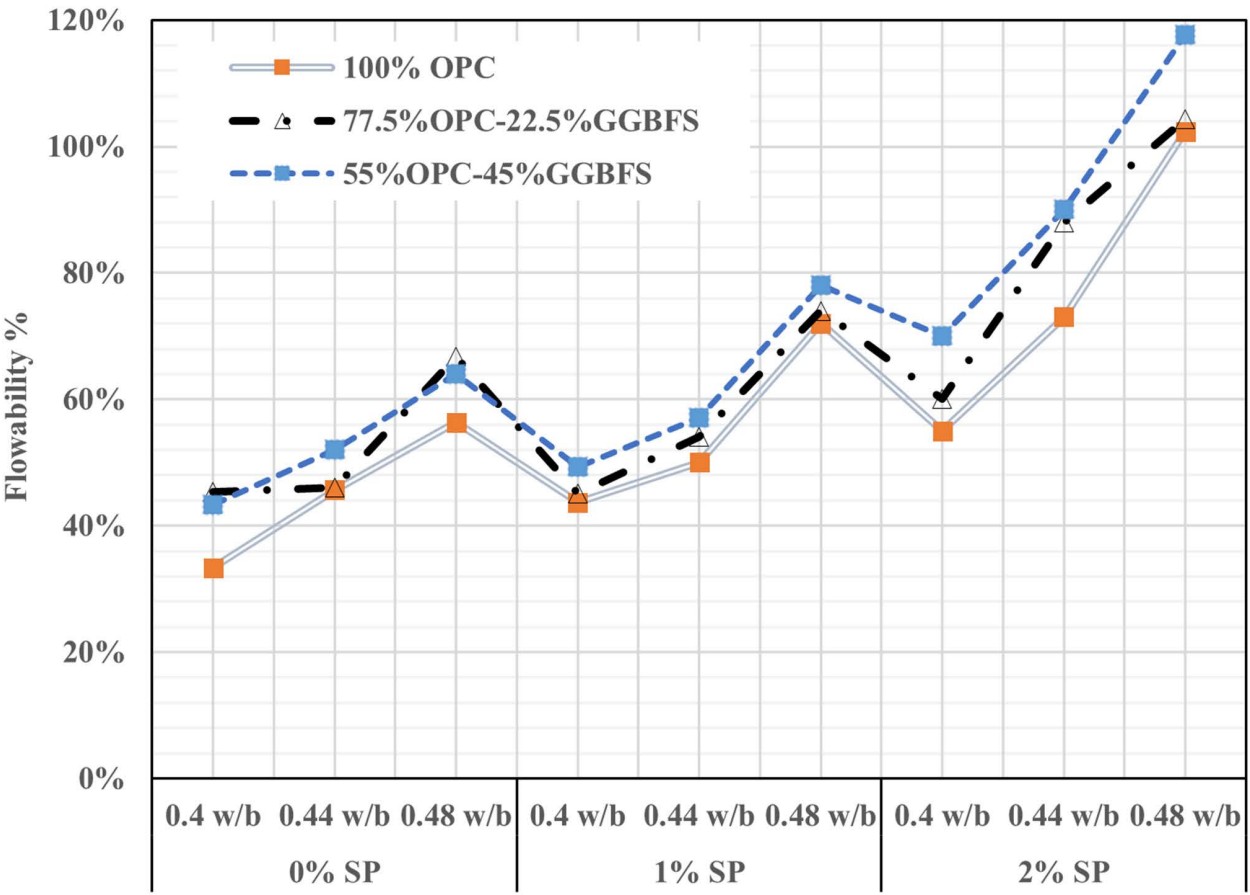

**Fig. 4. Flowability of cementitious mortar.**

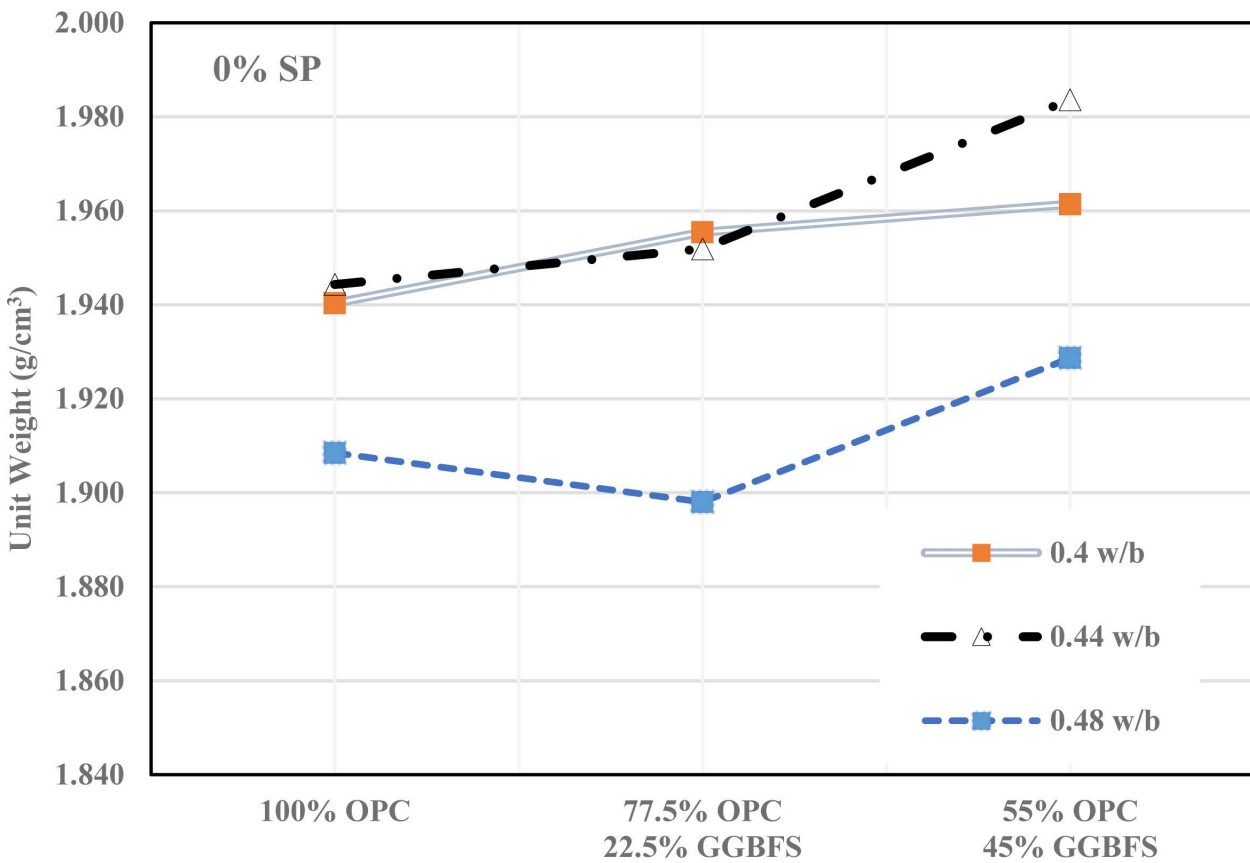

**Fig 5. Fresh density of a cementitious mortar without SP.**

mixes and the GGBFS replacement level are in inverse proportion. This adverse effect can be explained by the lower specific gravity of GGBFS compared to OPC. The specific gravity of the GGBFS used in this study was 2.9, while that of cement was 3.1. The lower specific gravity of GGBFS leads to a decrease in bulk density. In contrast, the addition of SP has a proportional effect on the bulk density in all mixes, regardless of GGBFS content and w/b ratios.

Furthermore, the correlation between the w/b and bulk density is intricate. A comprehensive analysis was conducted on samples characterized by different levels of SP and GGBFS replacement, uncovering a notable inverse relationship between bulk density and w/b ratio. This negative correlation was evident in samples devoid of SP at GGBFS replacement levels of 0% and 22.5%. Comparable outcomes were also noted in samples featuring 1% and 2% SP at GGBFS replacement levels of 22.5% and 45%, respectively. However, in contrast, bulk density and w/b ratio showed a positive correlation in pure cement blends containing 1% and 2% SP. Additionally, it was observed that when the replacement level of GGBFS was at 45%, with the addition of 0% and 1% SP, there was a noticeable impact on the bulk density. An increase in the w/b to 0.44 led to an increase in bulk density, however, a further increase to 0.48 resulted in a decrease in bulk density. The same trend was observed when the replacement level of GGBFS was at 22.5% with the addition of 2% SP. The discernible impact of SP becomes apparent upon its inclusion in the mixture, leading to an increased bulk density compared to formulations lacking SP. The effect of the SP in improving the packing of particles and

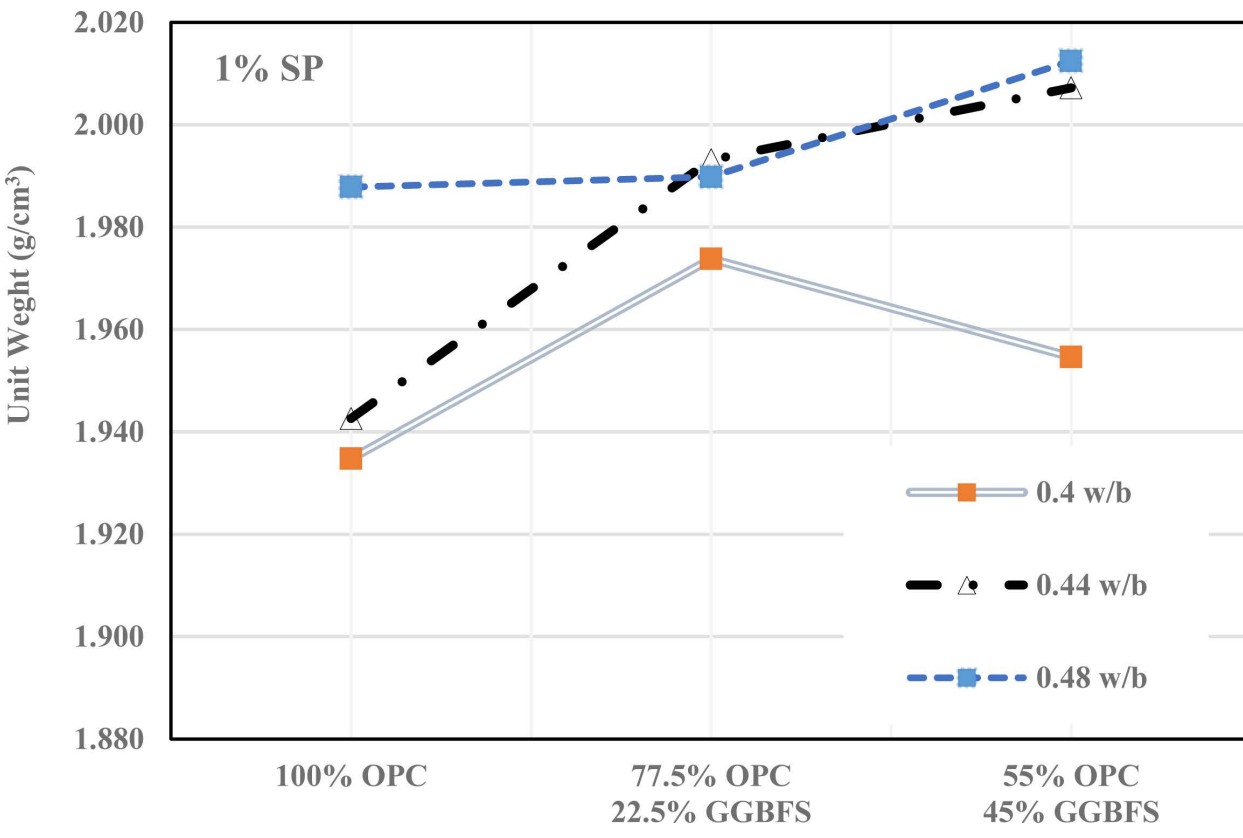

**Fig 6. Fresh density of cementitious mortar with 1% SP.**

providing better compaction inside the material has a significant impact on the substance's total bulk density.

### 3.3. Long-term compressive strength

The results of compressive strength at different GGBFS contents, w/b ratios, and SP contents at different ages (28, 56, 90, and 365 days) are illustrated in Fig 11 and Table 3. The compressive strength of concrete improves consistently with curing time showing notable gains at 56 and 90 days and continuing to increase uniformly across with different GGBFS percentages, w/b ratios, and SP concentration through 365 days. For instance, the highest percentage improvement in compressive strength at 56 days was observed at zero percent SP, or 34%, in specimens with full OPC at a w/b ratio of 0.4 demonstrated in Fig 11(a). The highest compressive strength increase (48.1%) at 90 days was observed in specimens with 2% SP. The result emphasizes the importance of curing time, SP concentration, and material composition on strength development. However, specimens with 22.5% GGBFS showed a considerable decrease in strength at 28 days compared to those with a 0.4 w/b ratio and different (SP) levels shown in Fig 11(b). At 56 and 90 days, respectively, these samples exhibited significant improvements in compressive strength of 57.2% and 67.9%. The particular combination of GGBFS content, w/b ratio, and SP concentration improved the growth of compressive strength in the specimens under examination, highlighting the influence of these variables on the material's long-term strength development. Similarly, as shown in Fig 11(c) Specimens with 45% GGBFS, 0.48 w/b ratio, and without SP dosage showed the highest compressive

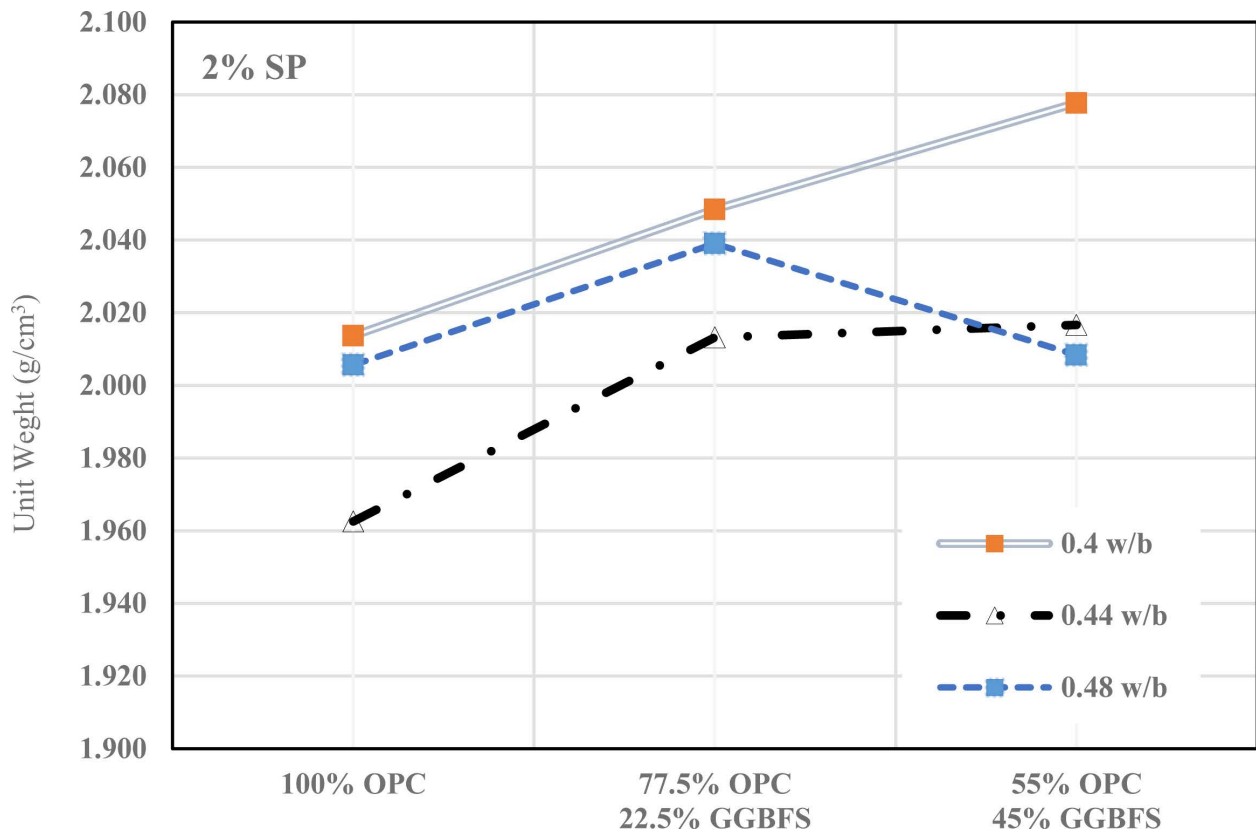

**Fig 7. Fresh density of cementitious mortar with 2% SP.**

strength improvements, with an increase of 78.2% at 56 days and 82% at 90 days, demonstrating the greatest strength gains relative to 28-day control values. Consistent with previous studies, Shubbar et al. [49] demonstrated that the addition of GGBFS to mortar decreased the compressive strength at early ages but ultimately improved after 28 days.

Specimens with 22.5% and 45% GGBFS show compressive strength gain by 90 days, surpassing 28-day OPC values, with an additional 10-12 MPa increase by 365 days. While higher GGBFS content initially reduces strength, it leads to greater long-term improvement. This underscores GGBFS's environmental benefits as a sustainable cement substitute. The improved performance of specimens with 22.5% and 45% GGBFS content is primarily due to the pozzolanic reaction, where GGBFS react with calcium hydroxide, forming additional calcium silicate hydrate (C-S-H) gel.

The findings also indicate that, in contrast to 56 to 90 days, the bulk of strength increase occurs between 28 and 56 days. With w/b = 0.4, 1% SP, and 0% GGBFS content, the peak compressive strength of 46.9 MPa at 90 days was 1.18 times higher than the 28-day compressive strength of 39.9 MPa. The superlative compressive strength development is 82% at the age of 90 days it has a remarkable rate of strength development 1.82 times greater than the compressive strength at 28 days (15.6 MPa), furthermore, the compressive strength slightly increases and 0.48 contains better compressive strength than 0.4 and 0.44 w/b. It can be concluded that extending the curing time to 90 days promoted the long-term compressive strength of all scenarios. After long-term curing in ambient conditions, the compressive strength of cement-based mortar slightly increased except for M1 decreased.

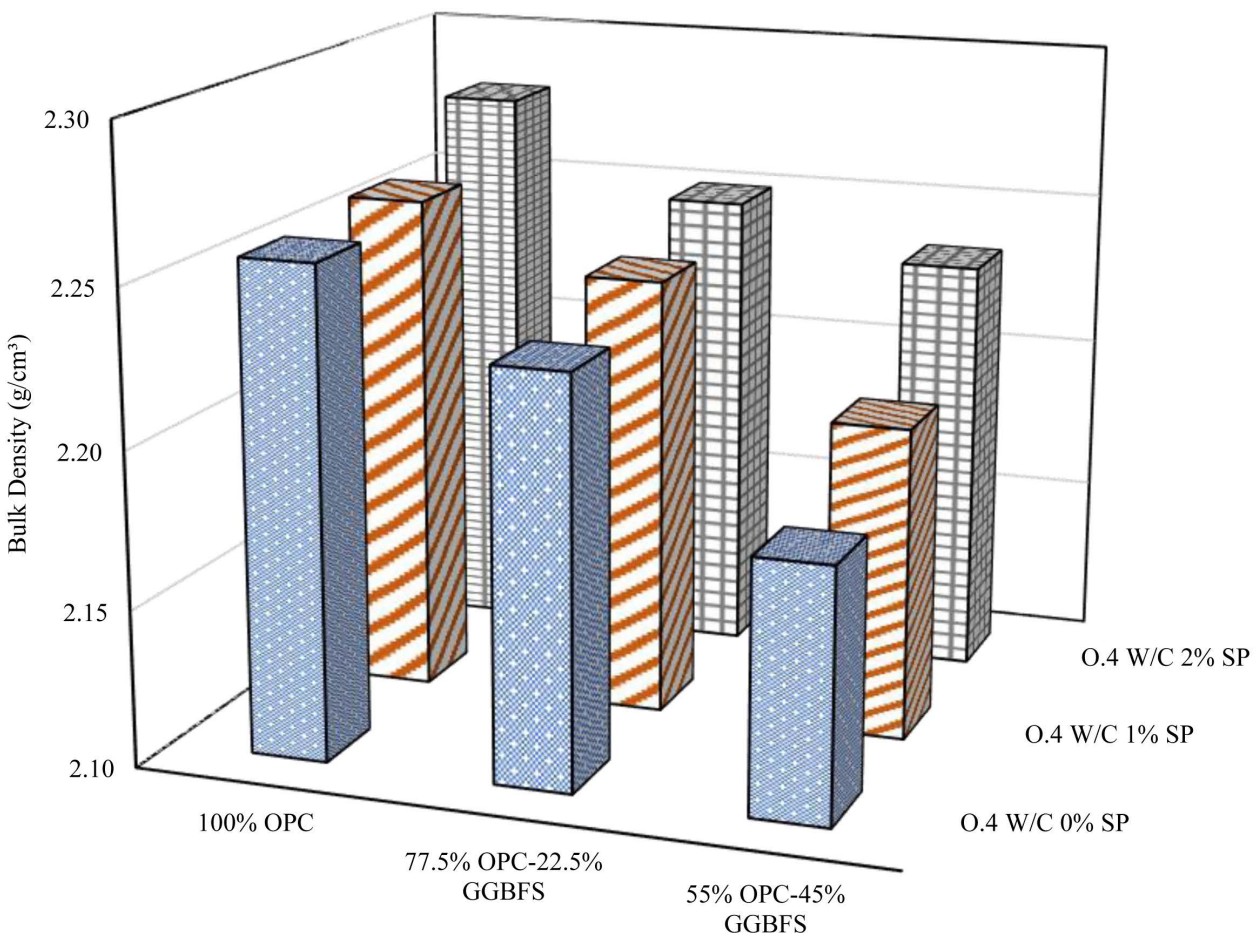

**Fig 8. Bulk density at 0.4 w/b.**

After a curing period of 365 days, the mixes containing GGBFS consistently demonstrated compressive strength values, indicating that the inclusion of GGBFS as a cement modifier does not adversely affect strength over time. The interaction between cement and GGBFS along with their distinct chemical compositions, appears beneficial for maintaining the integrity of the mixes. This finding underscores GGBFS potential as a sustainable alternative to traditional building materials. Notably, while mixes with 0% GGBFS yielded the highest compressive strength, partially replacing cement with GGBFS not only enhanced strength but also contributed to reducing $CO_2$ emission [50,51].

**Heating-cooling cycles of CS after 365 days.** Fig 12 presents a comprehensive overview of the experimental test samples evaluating the compressive strength of GGBFS-modified cement. The average outcome obtained from at least three samples is shown by each data point. Particularly, when compared to the same cycles carried out at 150°C, the compressive strength of fully cement binder specimens subjected to 30 cycles at both 50°C and 100°C showed an increase with rising w/b ratios and SP content, as shown in Fig 12(a) and confirmed through rigorous experimentation. This trend remained consistent irrespective of the sample temperature. The results show that cement-based mortar performs well at w/b ratios of 0.4 and that the compressive strength significantly increases at higher w/b ratios. It is interesting to note that the mixture with 1% SP performed better than the mixture with 2%

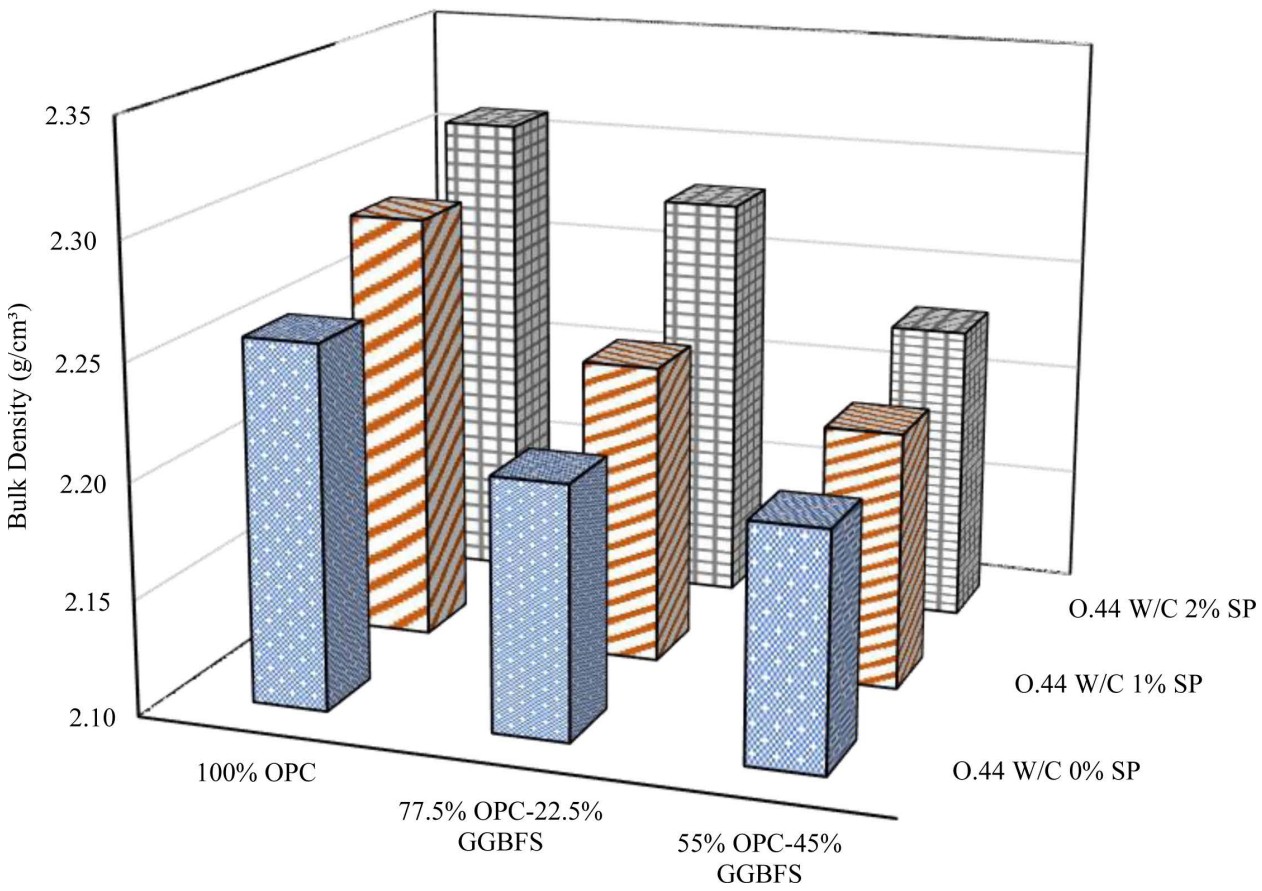

**Fig 9. Bulk density at 0.44 w/b.**

SP. In addition, the cement-based mortar's compressive strength showed an increasing trend following three different temperature exposures across thirty cooling cycles, highlighting the efficiency of a 0.4 w/b ratio in reaching greater levels of compressive strength. Following heating and cooling at temperatures of 50, 100, and 150 °C, the compressive strength increased. This experimental investigation proved that for construction building durability for a variety of seasons that occur in actuality, the long-term compressive strength of samples after exposure to varied climatic conditions showed the results for this experimental study.

The findings presented in Fig 12(b) and validated through experimentation reveal a distinct pattern: the compressive strength of the cement 77.5%-GGBFS 22.5% binder blend, subjected to 30 cycles at both 50 and 100 °C, displays a marked increase correlating with higher w/b ratios and SP concentrations. This improvement stands out notably when compared to samples subjected to 30 cycles at 150 °C. Consistent across the experiments is the evident correlation between elevated w/b ratios, augmented SP content, and enhanced compressive strength. Regardless of the samples' temperature, this was the case. According to the findings, the compressive strength of cement modified by GGBFS for 22.5%-based mortar showed acceptable performance but if the finding of cyclic exposure in temperature and cooling compared to control with 0.4 w/b ratio, and the strength increasing due to increasing temperature even for the SP have an advantage compressive strength and improving compressive strength by adding 1% of SP, the specimens contain mixes for 0.44 w/b ratio shows from figures M4,

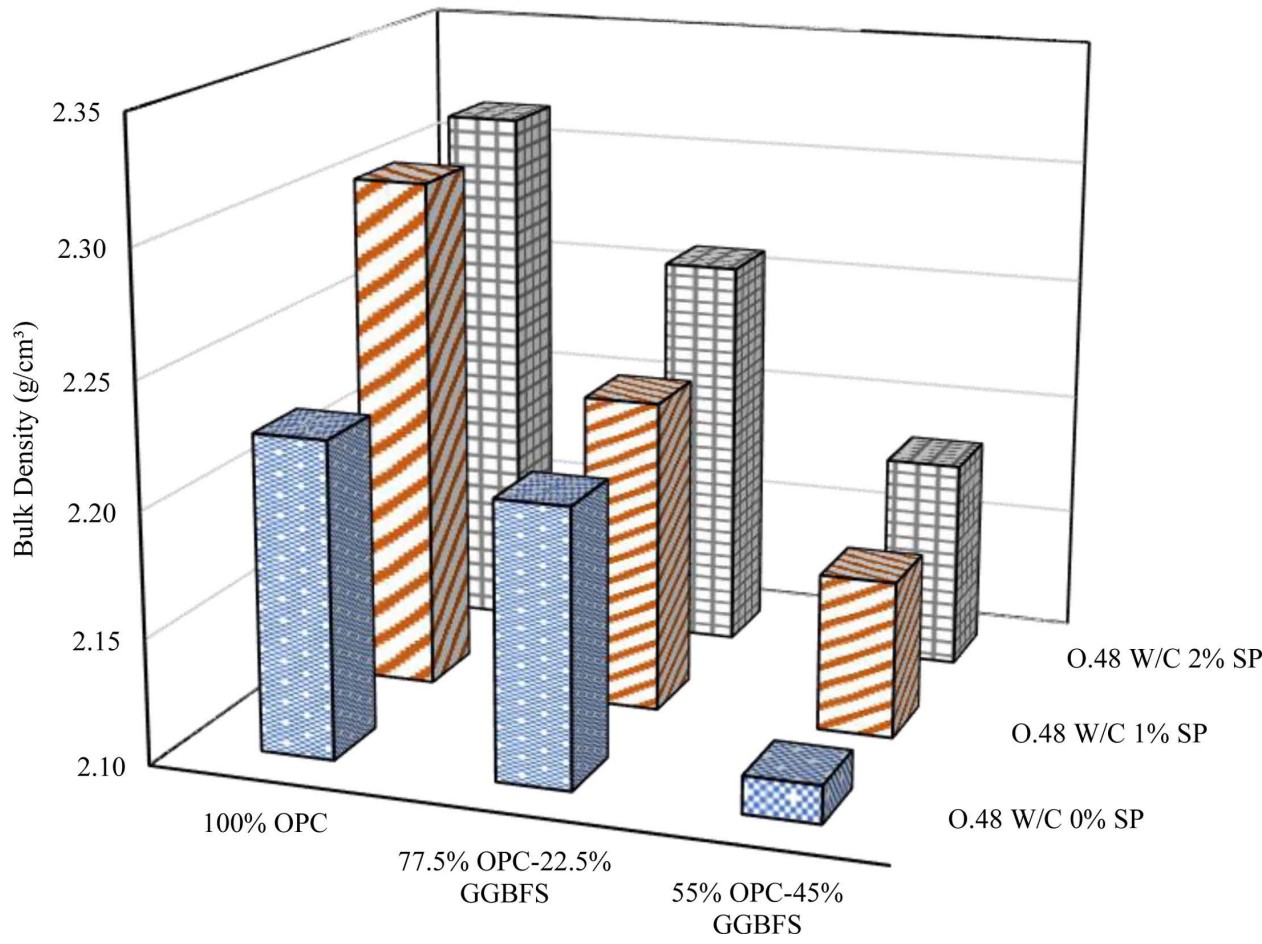

**Fig 10. Bulk density at 0.48 w/b.**

M5, and M6 higher Benefited results compared to 0.4 and 0.48 with contain 1% SP. Furthermore, regarding variations in compressive strength, the study found several interesting trends. Compressive strength and the w/b ratio were found to be positively correlated, with the w/b ratio indicating an obvious improvement. Certainly, the mix with 1% SP worked better than the one with 2% SP among the formulations tested, suggesting that there is a perfect concentration of SP for best results. Furthermore, after 30 heat cycles at three different temperatures, the addition of GGBFS, at a rate of about 22.5%, as a cement substitute continuously strengthened compressive strength. Significantly, a w/b ratio of 0.44 emerged as optimal, yielding the highest compressive strength. This enhancement persisted even after subjecting the samples to cyclic heating and cooling at 50, 100, and 150°C, further amplifying the compressive strength of the modified cement mortar. The compressive strength increased for all mixes when exposed to 50 and 100 °C but for 150 °C revised slightly decreasing. This experimental investigation proved that for construction building durability for a variety of seasons that occur in actuality, the long-term compressive strength of samples after exposure to varied climatic conditions showed the results for this experimental study. Specimens have been going to endure under exposed temperatures for 50 and 100 °C for even 30-time cyclic cooling and heating. The mechanical behavior under high temperature was better than that after high temperature, according to the researcher [52], who also noted that strength, elastic modulus, peak

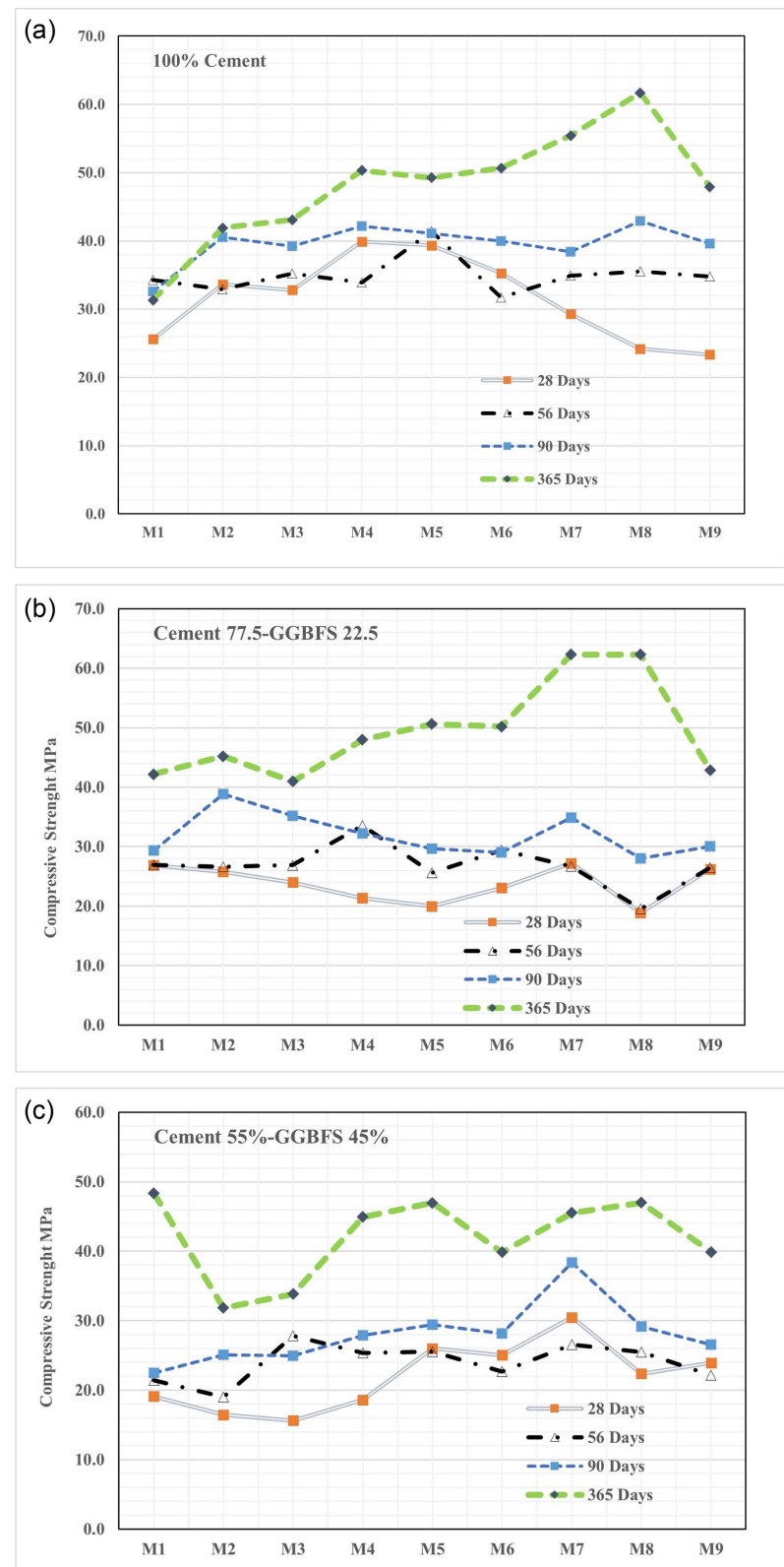

**Fig 11. (a). Long-term compressive strength development of fully cement. (b). Long-term compressive strength development of 22.5% GGBFS blended cement. (c). Long-term compressive strength development of 45% GGBFS blended cement.**

**Table 3. The Results of long-term, temp cycles of compressive strength and water absorption including different mixtures.**

| Mixtures | Groups | Control | Compressive Strength (MPa) - Temp Cycles | | | Compressive Strength (MPa) - Long term | | | | Water absorption (%) | |
|---|---|---|---|---|---|---|---|---|---|---|---|
| | | | 30 Cycle 50° | 30 Cycle 100° | 30 Cycle 150° | 28 d | 56 d | 90 d | 365 d | 28 d | 56 d |
| Cement 100% | M1 | 31.3 | 42.8 | 54.6 | 55.2 | 25.6 | 34.3 | 32.6 | 31.3 | 8.1 | 7.6 |
| | M2 | 41.9 | 51.5 | 53.5 | 46.3 | 33.6 | 32.9 | 40.5 | 41.9 | 8.2 | 7.8 |
| | M3 | 43.1 | 41.8 | 56.9 | 53.3 | 32.8 | 35.2 | 39.2 | 43.1 | 9.1 | 8.7 |
| | M4 | 50.3 | 63.4 | 75.9 | 63.4 | 39.9 | 33.9 | 42.1 | 50.3 | 7.0 | 6.7 |
| | M5 | 49.3 | 71.8 | 67.6 | 64.1 | 39.3 | 41.4 | 41.1 | 49.3 | 7.2 | 6.9 |
| | M6 | 50.7 | 58.0 | 63.1 | 58.2 | 35.2 | 31.7 | 40.0 | 50.7 | 8.1 | 7.8 |
| | M7 | 55.4 | 39.7 | 60.6 | 47.0 | 29.3 | 34.9 | 38.4 | 55.4 | 7.5 | 7.2 |
| | M8 | 61.7 | 58.9 | 40.1 | 46.1 | 24.2 | 35.5 | 42.9 | 61.7 | 7.6 | 7.2 |
| | M9 | 47.9 | 50.3 | 47.4 | 44.4 | 23.3 | 34.8 | 39.6 | 47.9 | 8.7 | 8.3 |
| Cement 77.5%-GGBFS 22.5% | M1 | 42.2 | 47.0 | 42.2 | 35.4 | 26.9 | 26.9 | 29.3 | 42.2 | 8.4 | 8.0 |
| | M2 | 45.2 | 33.6 | 38.9 | 40.2 | 25.8 | 26.6 | 38.9 | 45.2 | 9.8 | 9.0 |
| | M3 | 40.9 | 53.1 | 63.8 | 55.7 | 24.0 | 26.9 | 35.2 | 40.9 | 9.6 | 9.2 |
| | M4 | 48.0 | 50.9 | 62.5 | 59.0 | 21.3 | 33.5 | 32.2 | 48.0 | 7.9 | 7.5 |
| | M5 | 50.6 | 45.8 | 30.4 | 31.6 | 20.0 | 25.6 | 29.7 | 50.6 | 9.0 | 8.6 |
| | M6 | 50.2 | 34.1 | 54.0 | 55.0 | 23.1 | 29.4 | 29.0 | 50.2 | 9.4 | 9.1 |
| | M7 | 62.3 | 40.2 | 47.1 | 55.6 | 27.2 | 26.7 | 34.8 | 62.3 | 8.0 | 7.9 |
| | M8 | 62.3 | 44.4 | 45.7 | 47.5 | 18.9 | 19.5 | 28.0 | 62.3 | 8.4 | 8.2 |
| | M9 | 42.9 | 61.8 | 36.1 | 51.9 | 26.2 | 26.4 | 30.0 | 42.9 | 8.4 | 8.2 |
| Cement 55%-GGBFS 45% | M1 | 48.3 | 32.0 | 43.1 | 43.6 | 19.1 | 21.4 | 22.5 | 48.3 | 7.8 | 7.6 |
| | M2 | 31.8 | 35.8 | 33.2 | 32.5 | 16.4 | 19.0 | 25.1 | 31.8 | 8.7 | 8.4 |
| | M3 | 33.8 | 43.9 | 32.7 | 44.3 | 15.6 | 27.8 | 25.0 | 33.8 | 9.3 | 9.1 |
| | M4 | 44.9 | 55.5 | 42.9 | 48.4 | 18.6 | 25.3 | 27.9 | 44.9 | 8.4 | 8.0 |
| | M5 | 46.9 | 36.4 | 50.3 | 39.0 | 26.0 | 25.5 | 29.4 | 46.9 | 8.5 | 8.2 |
| | M6 | 39.9 | 39.8 | 48.0 | 41.0 | 25.0 | 22.7 | 28.2 | 39.9 | 8.7 | 8.4 |
| | M7 | 45.5 | 66.1 | 45.3 | 55.0 | 30.5 | 26.5 | 38.4 | 45.5 | 7.4 | 7.1 |
| | M8 | 47.0 | 40.2 | 44.0 | 54.0 | 22.3 | 25.5 | 29.2 | 47.0 | 8.7 | 8.5 |
| | M9 | 39.9 | 50.3 | 44.5 | 52.0 | 23.9 | 22.2 | 26.5 | 39.9 | 9.5 | 9.3 |

strain, and other properties degraded with temperature increases. There could be a number of reasons for the rise in compressive strength observed with increasing cycle temperatures [53]. Firstly, at higher temperatures, cementitious materials often undergo enhanced hydration reactions, leading to better bonding and increased strength within the material's matrix. Additionally, the thermal cycling might induce thermal stresses that could further densify the structure, resulting in improved strength properties.

The compressive strength of cement modified at temperatures of 50, 100, and 150 °C contains 55% of cement and GGBFS 45% binder with 30 cycling cycles, as shown in Fig 12(c) and confirmed by testing. The study findings reveal an intriguing trend in the compressive strength of cement modified by GGBFS within a 45%-based mortar composition. Initially, under cyclic exposure to varying temperatures and cooling, the compressive strength exhibited an unexpected decrease at 50°C when compared to the control sample with a 0.4 w/b ratio. This early decrease could be explained by possible modifications to the hydration mechanism or by modernization brought on by the colder temperature. As the temperature increased to 100°C and 150°C, the compressive strength did, however, indicate an increase.

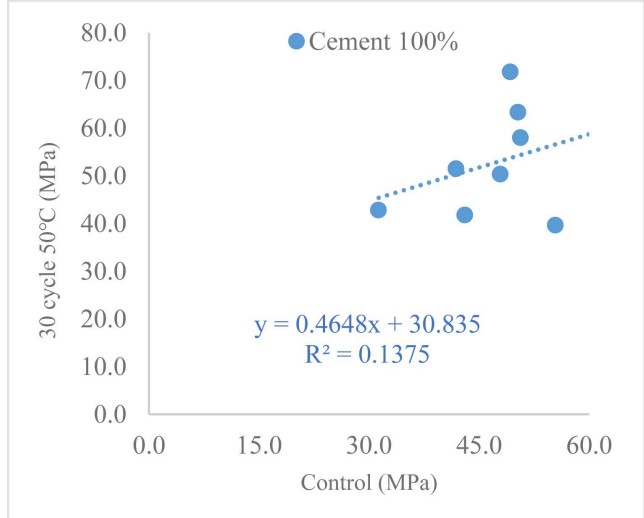

$y = 0.4648x + 30.835$
$R^2 = 0.1375$

$y = -0.1139x + 63.201$
$R^2 = 0.0085$

| Groups | Control | 30 Cycle 50° | 30 Cycle 100° | 30 Cycle 150° |
|--------|---------|--------------|---------------|---------------|
| M1 | 31.3 | 42.8 | 54.6 | 55.2 |
| M2 | 41.9 | 51.5 | 53.5 | 46.3 |
| M3 | 43.1 | 41.8 | 56.9 | 53.3 |
| M4 | 50.3 | 63.4 | 75.9 | 63.4 |
| M5 | 49.3 | 71.8 | 67.6 | 64.1 |
| M6 | 50.7 | 58.0 | 63.1 | 58.2 |
| M7 | 55.4 | 39.7 | 60.6 | 47.0 |
| M8 | 61.7 | 58.9 | 40.1 | 46.1 |
| M9 | 47.9 | 50.3 | 47.4 | 44.4 |

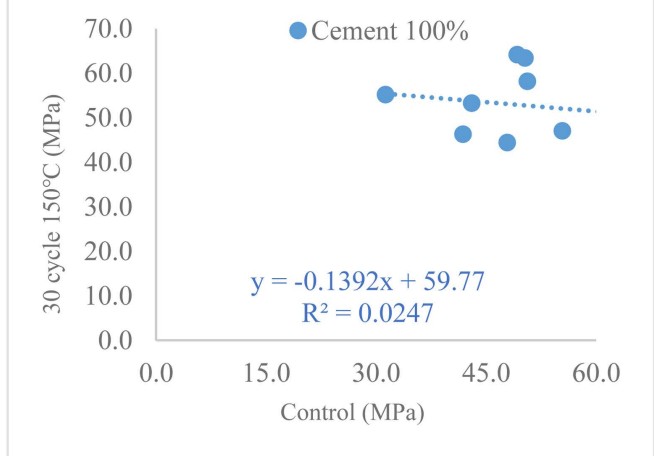

$y = -0.1392x + 59.77$
$R^2 = 0.0247$

The strength that has been shown to increase with temperature may be the result of improved chemical reactions, stronger particle binding, or a decrease in surface morphology weak points that were created during the lower temperature period. It is clear that high ranges of GGBFS as a binder do not contain SP to increase compressive strength since specimens without SP perform better than specimens with SP. The specimens with mixes for 0.48 w/b ratio exhibit increased strength in figures M7, M8, and M9 compared to 0.4 and 0.44 without SP, and the strength marginally decreases when 1 and 2 percent of SP is added. Additionally, after being exposed to three different temperatures over the course of 30 cycles of cooling, the compressive strength of cement modified by GGBFS that was around 45% replaced to cement mortar increased, and a ratio of 0.48 w/b was discovered to achieve a higher level of compressive strength. Since GGBFS is an iron substance that absorbs heat and strengthens bonds, it is indicated that GGBFS materials could resist temperature and higher temperatures might excessively heat producing and faster hydration process. The compressive strength increases for all mixtures when exposed to 150 °C. This experimental study demonstrated that for construction building durability for a variety of seasons that actually occur, even

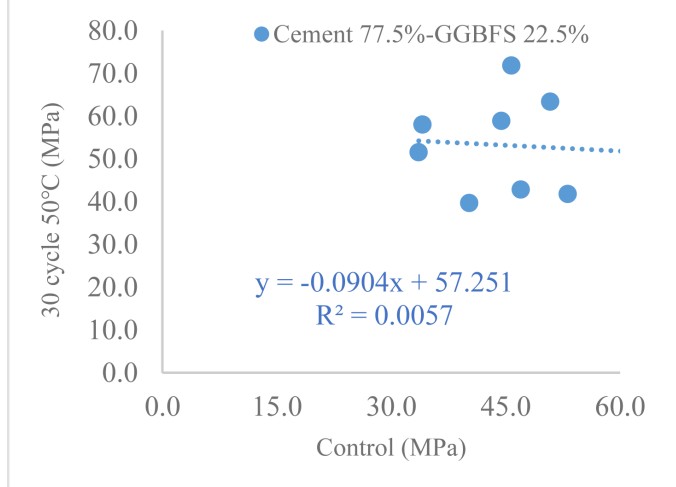

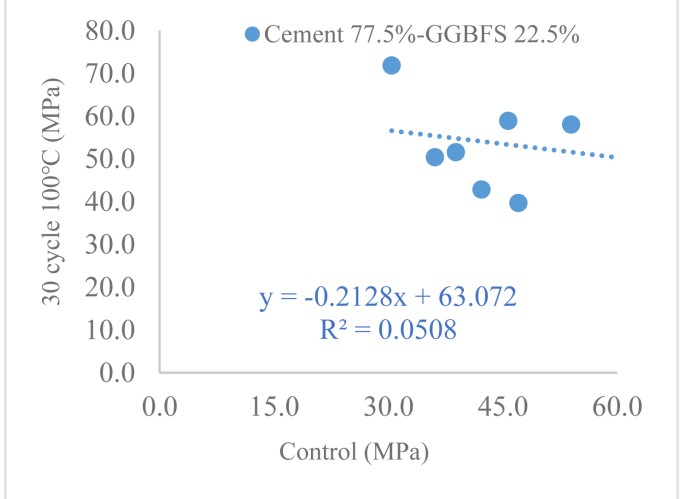

| Groups | Control | 30 Cycle 50° | 30 Cycle 100° | 30 Cycle 150° |
|--------|---------|--------------|---------------|---------------|
| M1 | 42.2 | 47.0 | 42.2 | 35.4 |
| M2 | 45.2 | 33.6 | 38.9 | 40.2 |
| M3 | 40.9 | 53.1 | 63.8 | 55.7 |
| M4 | 48.0 | 50.9 | 62.5 | 59.0 |
| M5 | 50.6 | 45.8 | 30.4 | 31.6 |
| M6 | 50.2 | 34.1 | 54.0 | 55.0 |
| M7 | 62.3 | 40.2 | 47.1 | 55.6 |
| M8 | 62.3 | 44.4 | 45.7 | 47.5 |
| M9 | 42.9 | 61.8 | 36.1 | 51.9 |

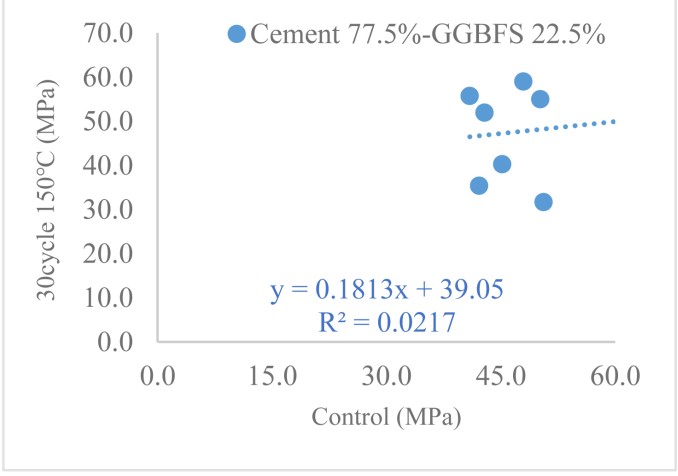

for high-range GGBFS that demonstrated the results for this experimental study, samples' long-term compressive strength after exposure to varied climatic conditions was significant. Specimens are going to be subjected to temperatures of 50, 100, and 150 ° C for even 30 cycles each of chilling and heating. Concrete mixtures made with Portland cement alone had lower compressive casting and curing at +42C than those made with slag as a partial replacement of cement (up to 40%) [54].

## 3.5. Weight loss

The weights of the specimens underwent significant variations after 30 cycles of heating and cooling. The specimens lost weight as the temperature rose from 50 to 150 ° C. The evaporation of moisture, the release of volatile substances, and probable chemical reactions taking place inside the material are just a few of the causes of this weight loss. The weight loss process was accelerated at temperatures of 100°C and 150°C, indicating a greater amount of moisture evaporation and volatile chemical release. These results imply that the specimens lost moisture and volatile components due to the cyclic thermal loading and high temperatures, which

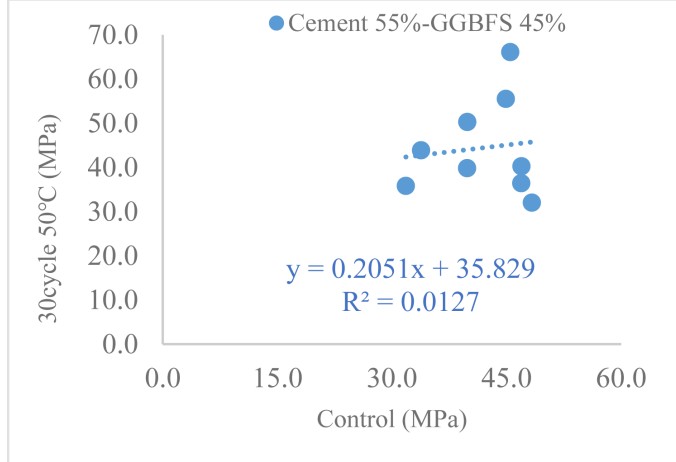

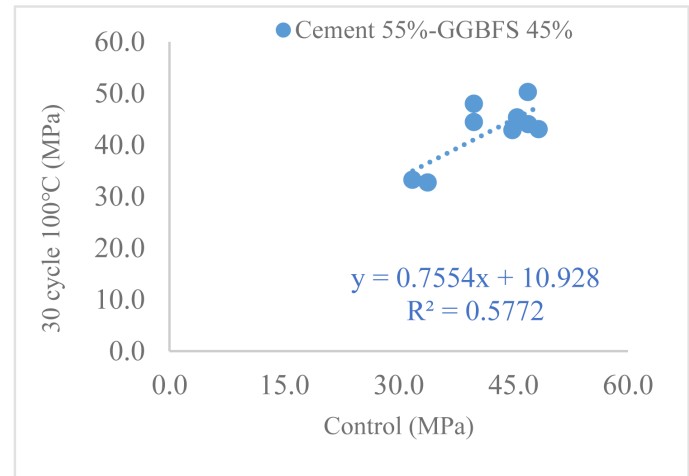

| Groups | Control | 30 Cycle 50° | 30 Cycle 100° | 30 Cycle 150° |
|--------|---------|--------------|---------------|---------------|
| M1 | 48.3 | 32.0 | 43.1 | 43.6 |
| M2 | 31.8 | 35.8 | 33.2 | 32.5 |
| M3 | 33.8 | 43.9 | 32.7 | 44.3 |
| M4 | 44.9 | 55.5 | 42.9 | 48.4 |
| M5 | 46.9 | 36.4 | 50.3 | 39.0 |
| M6 | 39.9 | 39.8 | 48.0 | 41.0 |
| M7 | 45.5 | 66.1 | 45.3 | 55.0 |
| M8 | 47.0 | 40.2 | 44.0 | 54.0 |
| M9 | 39.9 | 50.3 | 44.5 | 52.0 |

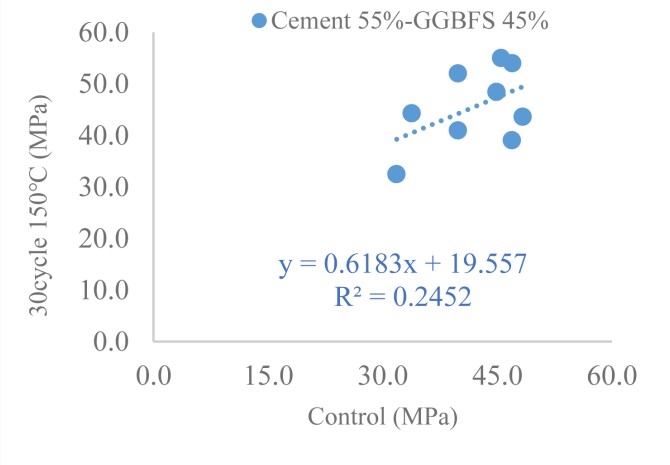

**Fig 12. (a). Compressive strength of fully cement under heating and cooling procedure.** (b). Compressive strength of cement modified of 22.5% GGBS under heating and cooling procedure. (c). Compressive strength of cement was modified by about 45% based evaluated under heating and cooling procedure.

would have affected their physical characteristics and overall performance. Fig 13 demonstrates the percentages of losing weight of specimens after heating and cooling cyclic after different temperatures.

Fig 13(a) demonstrates the result of mortar with full cement. When subjected to a temperature of 50°C, the specimens comprised of 100% cement exhibited weight loss that varied depending on the w/b ratio. For specimens with a w/b ratio of 0.4 and 0.44, the weight loss was approximately 2%. This can be attributed to the evaporation of moisture within the cement matrix due to the elevated temperature. When exposed to a higher temperature of 100°C, the specimens composed solely of 100% cement experienced more significant weight loss compared to the lower temperature scenario. For specimens with a w/b ratio of 0.4 and 0.44, the weight loss ranged between 4% and 5%. This substantial weight loss can be attributed to the intensified evaporation of moisture within the cement

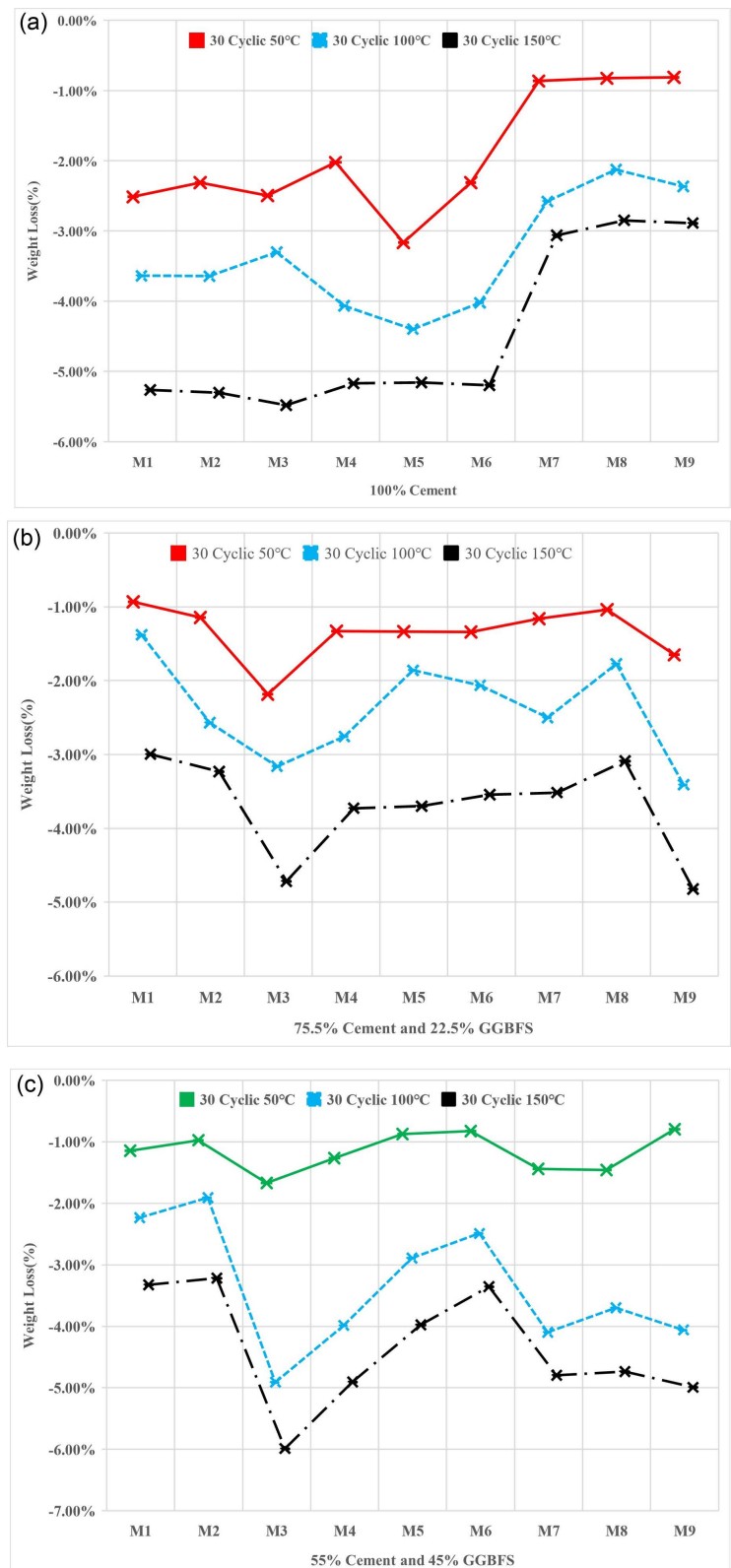

**Fig 13. (a). Weight loss percentages of 100% cement binder specimens after heating and cooling process.**
(b). Weight loss percentages of 75.5% cement and 22.5% GGBFS binder specimens after heating and cooling process.
(c). weight loss percentages of 55% cement and 45% GGBFS binder specimens after heating and cooling process.

matrix due to the elevated temperature. Furthermore, for specimens with a higher w/b ratio of 0.48, the weight loss increased to approximately 2% to 3% more than the lower w/b ratios. And for 150°C, specimens with a w/b ratio of 0.4 and 0.44, the weight loss ranged between 5% and 6%. Additionally, for specimens with a higher w/b ratio of 0.48, the weight loss increased by approximately 3%. However, for specimens with a higher w/b ratio of 0.48, the weight loss increased to approximately 1% more than the lower w/b ratios. This suggests that a higher water content in the mix resulted in slightly greater evaporation and subsequent weight loss.

Fig 13(b) The weight loss was seen to be rather minimal when the specimens were exposed to temperatures of 50°C and included a mixture of 75.5% cement and 22.5% GGBFS. The weight loss ranged from 1.2 to 2.2% for specimens having a w/b ratio of 0.4. The high temperature caused moisture to evaporate from the cementitious matrix, resulting in this slight weight loss. In a similar vein, specimens exhibiting a w/b ratio of 0.44 saw a weight loss of approximately 1.5%. Additionally, specimens with a higher w/b ratio of 0.48 showed weight loss ranging from 1.1 to 1.8%. These results showed that the addition of GGBFS to the cement matrix marginally reduced weight loss when compared to specimens made completely of cement. A weight loss that was noticeable was seen at 100°C. Weight loss for specimens with a w/b ratio of 0.4 ranged from 1.3 to 3.2%. Similar to this, specimens with a w/b ratio of 0.44 lost weight in the range of 0.8 to 2.1%. Additionally, specimens M7 and M8 showed weight losses of 1.8 and 4.3%, respectively, when comparing various mix forms. Weight loss for specimens with a w/b ratio of 0.48 was about 3.3%. There was a noticeable weight loss at 150 °C. The weight loss ranged from 3.1 percent to 4.8 percent for specimens having a w/b ratio of 0.4. The same was true for specimens with a w/b ratio of 0.44, who lost weight by about 3.7%. Additionally, specimens M7 and M8 showed weight losses of 3.1 and 3.5%, respectively, when comparing various mix patterns. Weight loss for specimens with a w/b ratio of 0.48 was about 4.9%. These findings suggest that weight loss continued to occur at 100°C despite the inclusion of GGBFS in the cement mixture. This result emphasized the impact of taking material behavior and suitable mixture designs when exposed to high temperatures into consideration. The weight loss observed suggests that the specimens experienced significant evaporation and potential chemical reactions, according to this finding.

A substantial weight loss was seen when the specimens which contained 45% GGBFS and 55% cement were heated and cooled 30 times at 150 °C shown in Fig 13(c). The weight loss ranged from about 1.2% to 1.8% for specimens with a w/b ratio of 0.4. Similar to this, specimens with a w/b ratio of 0.44 lost between 0.8 and 1.2% of their body weight. Additionally, both specimens M7 and M8 showed a weight reduction of roughly 1.4% when comparing various mix patterns. M8 lost 1.2%, but M7 lost slightly more, 1.8%. Weight loss for specimens with a w/b ratio of 0.48 was about 4.9%. These results imply that GGBFS, especially at lower w/b ratios, provides some resistance to weight loss at 150 °C in the cement mixture. However, the weight loss was more pronounced at larger w/b ratios. When developing and using cement mixes with GGBFS in applications exposed to high temperatures, it is imperative to take these weight loss effects into account.

### 3.6. Prediction and actual compressive strength

In this study, cement was modified using GGBFS, and the effect of different w/b ratios was evaluated Fig 14. To determine the relationship between the chosen parameters and their corresponding outcomes, the Design Expert software was utilized with the central composite design (CCD) method. The CCD method involves a combination of factorial, axial, and center points, allowing for the identification of both the predicted and actual results. The analysis

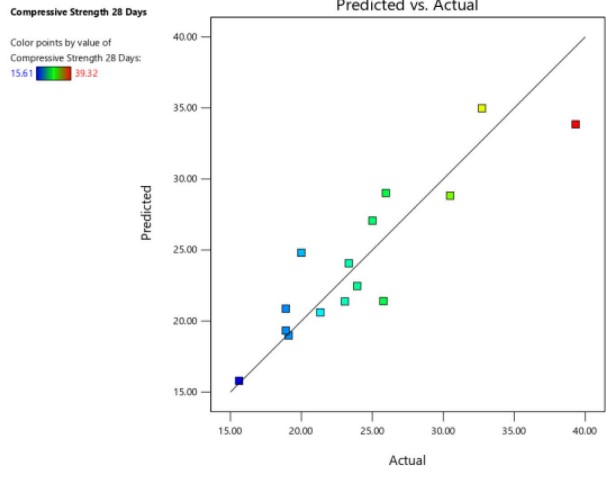

Prediction vs Actual Result at 28 days

Prediction vs Actual Result at 56 days

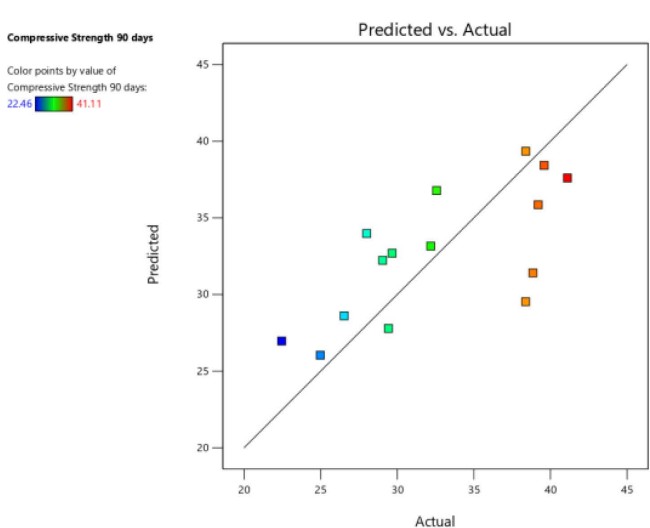

Prediction vs Actual Result at 90 days

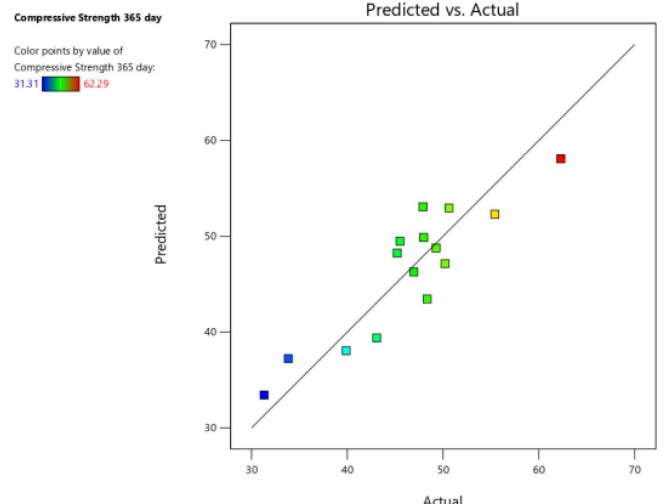

Prediction vs Actual Result at 365 days

**Fig 14. Prediction and actual results for compressive strength.**

of the predicted results compared to the actual results revealed a close alignment between the two, indicating a high level of accuracy in the study. This indicates that the chosen w/b ratios, as evaluated using the central composite design method, were effective in predicting the outcome of the modified cement. The proximity of the predicted and actual results suggests that the experimental design was successful in capturing the important factors influencing the cement modification process. These findings further validate the significance of the study and highlight the reliability of the central composite design method for evaluating and predicting the effects of GGBFS on cement properties. Response Surface Methodology (RSM) has been used for statistical analysis and it has been mostly used in the field of concrete technology for rapid and reliable prediction of the properties of concrete [55,56].

### 3.7. Durability performance

**Water absorption.** The water absorption at different GGBFS contents, w/b ratios, and SP dosages is shown in Figs 15–17. The involving interactions of GGBFS content, w/b ratios, and SP concentrations can be responsible for the observed variations in water absorption. When 22.5% GGBFS is substituted for cement at a 0.4 w/b ratio, there may be an initial increase in water absorption due to the larger porosity or increased permeability that this substitution provides demonstrated from Fig 15. Increased water intrusion into the material could result from this modification. On the other hand, the following decrease in water absorption after adding a further 45% GGBFS replacement may indicate that the material has a number of filled or that the pore structure has improved, which will lower the water permeability.

At a 0.44 w/b ratio, one possible mechanism for the rise and subsequent reduction in water absorption with increasing GGBFS content is demonstrated in Fig 16. The GGBFS substitution may have caused the initial increase by introducing more porous components. Higher GGBFS levels, on the other hand, would indicate a microstructure improvement that lowers pore connectivity and complicates water inclusion. Moreover, the decline in water absorption was followed by an increase concerning GGBFS replacements.

A mixture with a 0.48 w/b ratio demonstrated in Fig 17, may suggest a complicated equilibrium between the changing microstructure and porosity. The first decrease may indicate better packing and decreased porosity, but an increase at a greater GGBFS concentration could indicate changes in the pore size distribution or connectivity, which could result in more water becoming absorbed.

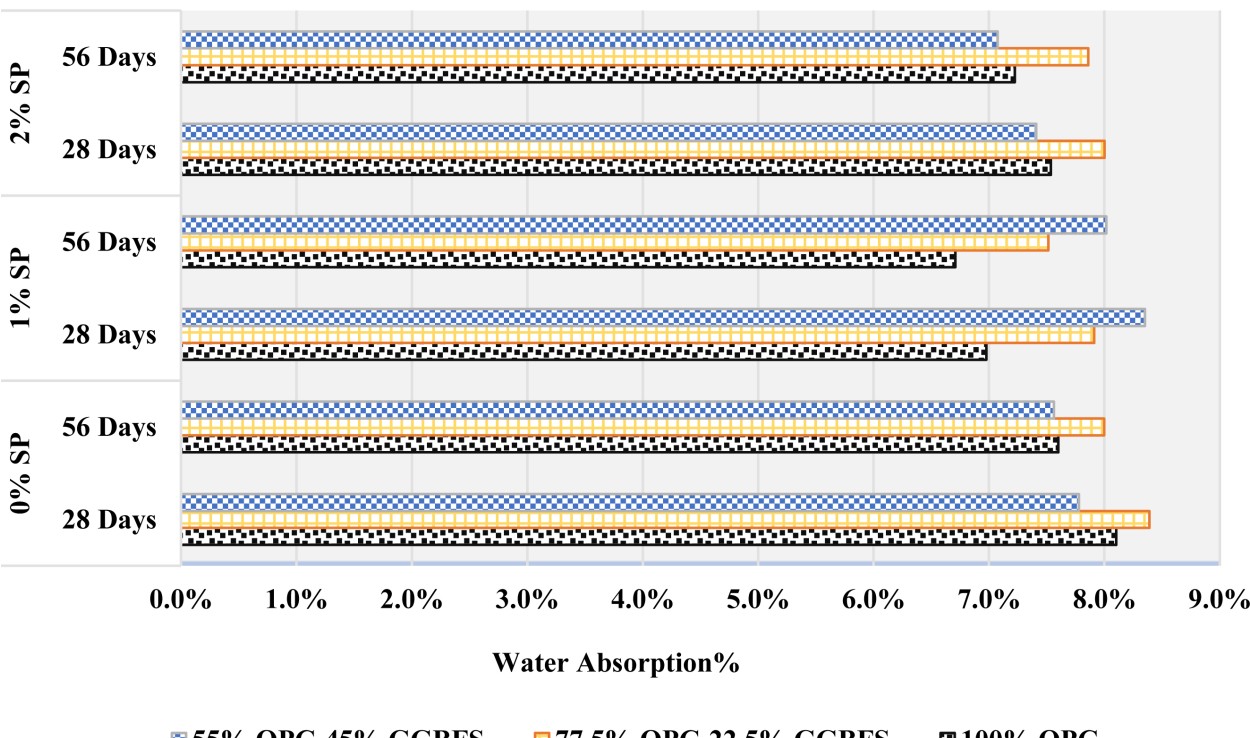

**Fig 15. The rate of absorption with 0.4 w/b.**

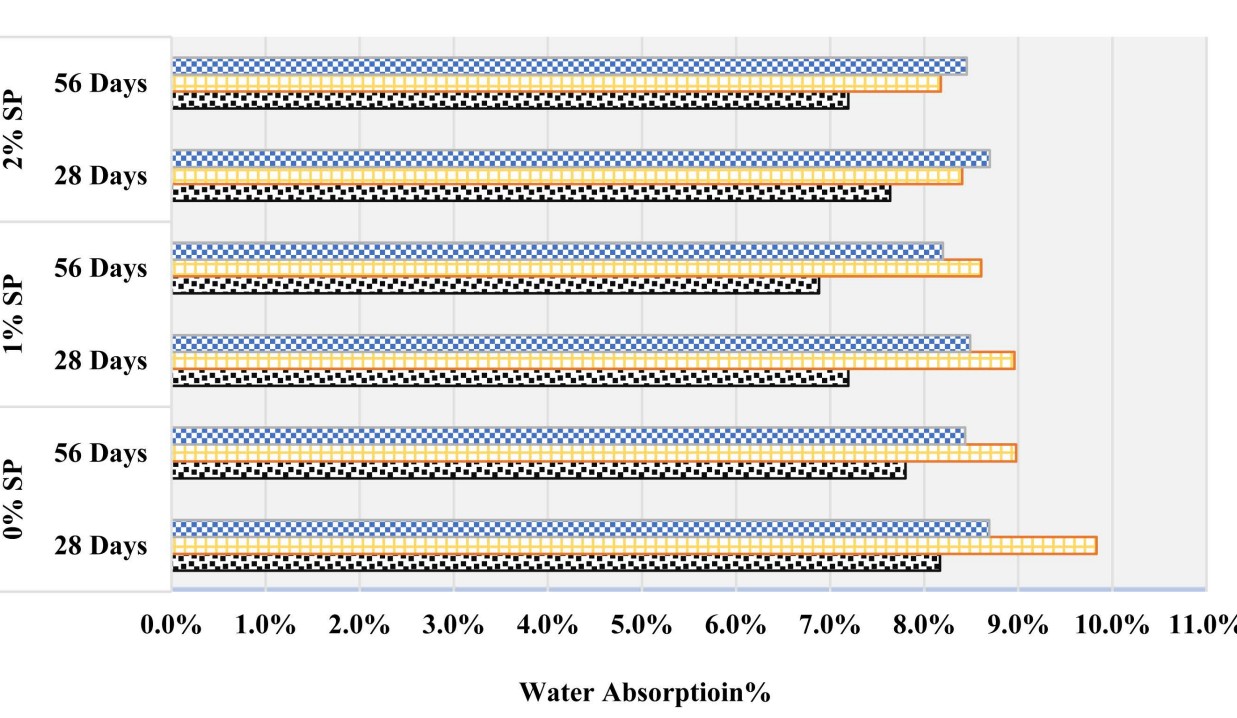

**Fig 16. The rate of absorption with 0.44 w/b.**

The material's porosity, pore structure, and integration are all affected by changes in GGBFS content, as well as the particular w/b ratio and SP concentration used in the mix. This is suggested by the varying water absorption behavior observed across different GGBFS replacements and w/b ratios. However, several variables can affect the link between water absorption and SP level, making it less clear. Higher SP content was associated with reduced water absorption at 0.4 and 0.44 w/b ratios when 22.5% GGBFS was substituted for cement. However, with pure cement mixes, the water absorption was initially reduced with the addition of 1% SP but increased with further increment in the SP content.

0.48 w/b with 45% GGBFS replacement showed the similar behavior. 45% GGBFS replacement mixes showed the opposite effect at w/b ratios of 0.4 and 0.44, where a higher SP content of 1% was associated with higher water absorption, while a higher SP content of 2% was associated with lower water absorption. This behavior was also observed in mixes with 0% and 22.5% GGBFS replacements at a 0.48 w/b. Furthermore, it can be seen, by comparing Fig 15–17, that as the w/b increases, the water absorption also increases in all mixes. This study demonstrates that an increase in the water content of cement leads to a heightened absorption rate. This phenomenon is attributed to the fact that a higher w/b ratio produces more porous mortar with larger capillary pores, which enables it to absorb greater quantities of water. It is worth mentioning that the behavior described above remained consistent at both 28 and 56 days. However, it is noteworthy that the rate of absorption showed a slight decrease at 56 days when compared to 28 days.

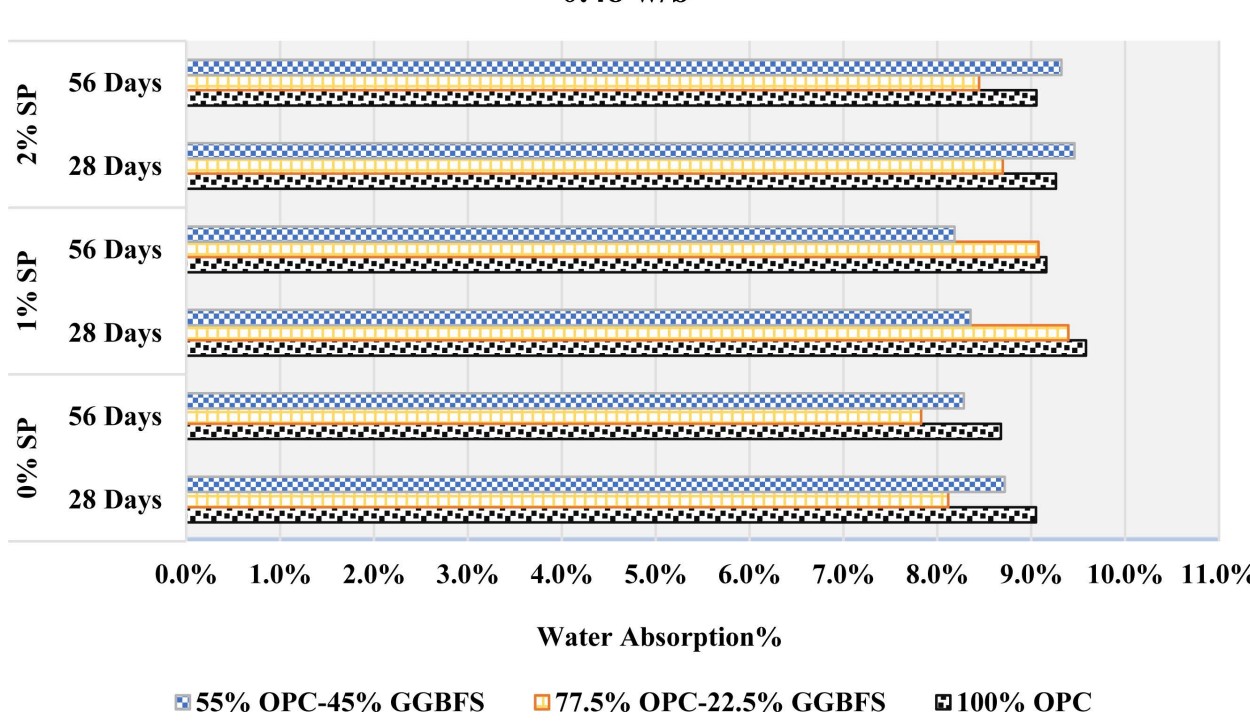

**Fig 17. The rate of absorption with 0.48 w/b.**

### 3.8. Microstructural analysis utilizing SEM

Scanning Electron Microscopy (SEM) has been utilized to analyze the cement modified with different GGBFS replacement levels. The three phases of binder analyzed specimens which are (100% cement, 22.5% GGBFS 77.5% cement, and 45% GGBFS and 55% cement) have been exanimated. Fig 18 demonstrates how the fully-cement specimens indicated the amount of GGBFS, portlandite (CH), and calcium silicate hydrate (C-S-H) pores, as well as the number of reached and unreached GGBFS microcracks. In the investigations conducted at room temperature, the microstructure was stabilized by the hydration process and reacting and bonding binder. In Fig 18 conducted that 22.5% and 45% of GGBFS binders were demonstrated, and when GGBFS was added to the binder the particles showed an interface quite homogeneous compared to the specimens because of the amorphous shapes of combined both cement and GGBFS. The number of pores decreased in the specimens containing GGBFS but the number of voids between aggregate and binders increased. The calcium silicate hydrate (C-S-H) crystals were explained and demonstrated in the 20-micrometer figures as having formed as a result of the pozzolanic process. The microstructure, which demonstrates a lot of calcium silicate hydrate crystals and is relatively denser after 28 days, results in high strength. In comparison to binders substituted with GGBFS to 22.5% and 45% compact with fewer pores, the microstructure of fully cement binders gave the calcium oxide in the matrix. OPC, the most widely used form of cement in building A mixed cement is what is produced by this mixture. When coupled with cement (OPC), GGBFS can partially replace cement. The calcium oxide in GGBFS and the calcium oxide in OPC help to create the calcium-silicate-hydrate gel (C-S-H gel), the primary binder in concrete, which is a solid and long-lasting gel. To give the concrete its strength and stability, this gel joins the cement and aggregate particles together. The high

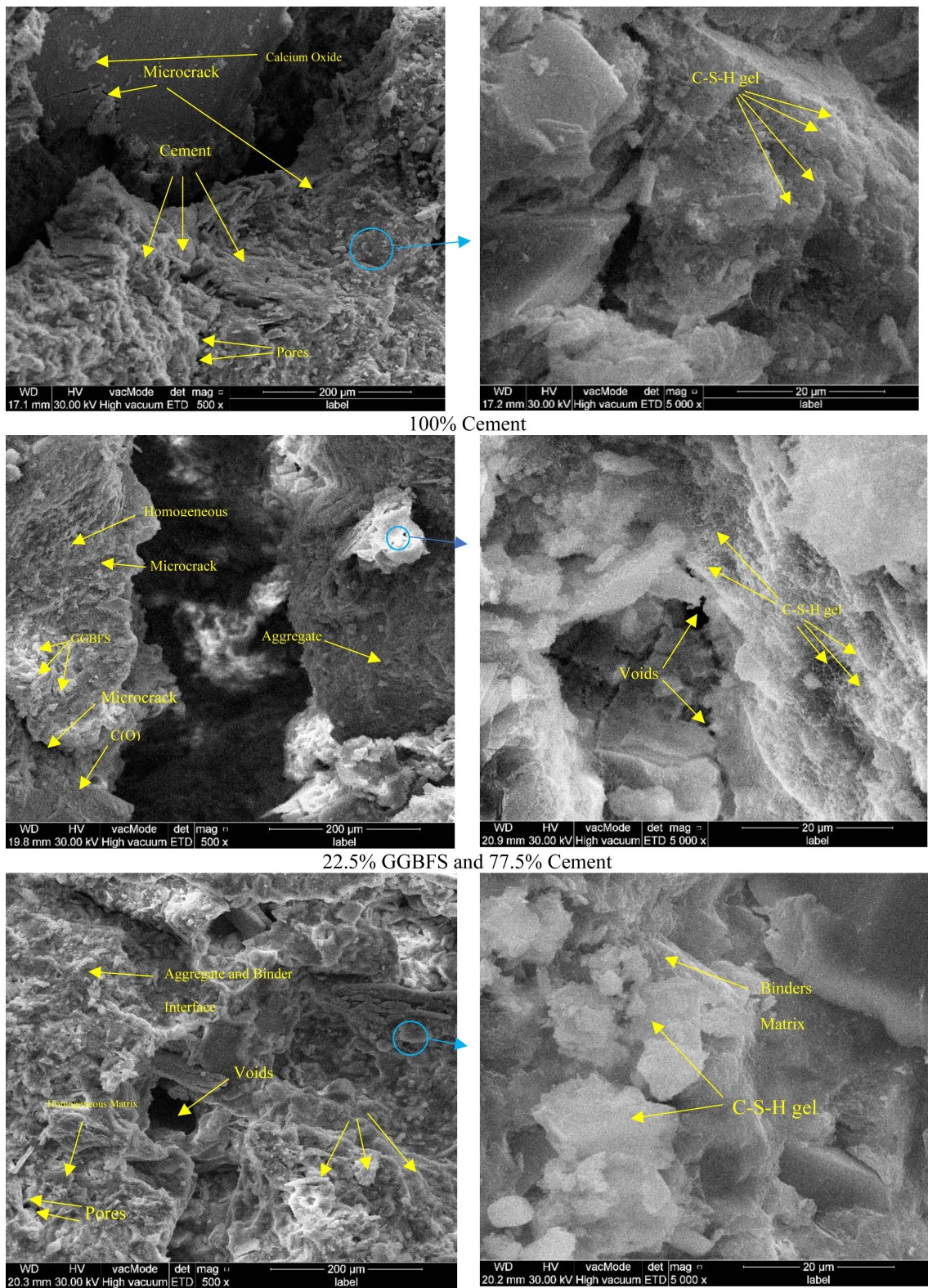

**Fig 18. SEM analysis of cement and GGBFS composite.**

concentration of calcium oxide, often known as lime or calcium oxide, in GGBFS, is one of its important properties. Due to its reaction with water to produce calcium hydroxide, often known as hydration, calcium oxide is a crucial component in the cementing process. The hardening and setting of concrete depend on this hydration process.

The significant consumption of calcium oxide demonstrates the beginning of the pozzolanic reaction, which accounts for the enhanced strength. The hydrated grains in the specimens with GGBFS are a result of hydration reactions that result in the development of C-S-H gel. Although there is little obvious interlocking between the hydrated grains, the overall microstructure is highly dense. The void intervals that exist between the hydrated particles are clearly visible. The increase in strength is caused by the development of these hydrated particles. The microstructure becomes denser in the presence of finely ground pozzolanic material like GGBFS, increasing the compressive strength [57].

## 4. Conclusions

This study evaluated the impact of varying water-to-binder (w/b) ratios and superplasticizer (SP) percentages in cementitious mortars, focusing on substituting Ground Granulated Blast Furnace Slag (GGBFS) for Ordinary Portland Cement (OPC). The investigation covered fresh and long-term compressive strength, thermal performance, microstructural properties, and water absorption. Key findings are as follows:

- Increasing GGBFS content, combined with higher SP dosages, significantly improved flowability, achieving a maximum flow rate of 118% with 45% GGBFS replacement, 2% SP, and a w/b ratio of 0.4. This highlights GGBFS's suitability for enhancing workability in construction.

- GGBFS replacement reduced carbon dioxide emissions, offering a sustainable alternative to OPC and supporting eco-friendly concrete production.

- GGBFS slightly reduced early compressive strength but showed significant long-term strength gains, particularly at 90 and 365 days, due to dense calcium silicate hydrate (C-S-H) gel formation.

- GGBFS mixtures exhibited resilience under heating and cooling cycles, indicating enhanced durability for structures in variable thermal environments.

- While higher w/b ratios increased water absorption, SP reduced it at specific ratios. GGBFS content influenced absorption variably, providing design flexibility for durability.

- Statistical analysis confirmed the reliability of the experimental results, with significant P-values reinforcing the validity of the observed trends in compressive strength and durability parameters.

These findings underscore the potential of GGBFS to enhance the sustainability and performance of concrete. By improving durability, reducing environmental impact, and maintaining long-term strength, GGBFS aligns with the construction industry's goals of sustainability and resilience.

## Author contributions

**Conceptualization:** Mahmood Hunar Dheyaaldin.

**Data curation:** Mahmood Hunar Dheyaaldin.

**Formal analysis:** Mahmood Hunar Dheyaaldin.

**Investigation:** Mahmood Hunar Dheyaaldin.

**Methodology:** Mahmood Hunar Dheyaaldin.

**Software:** Mahmood Hunar Dheyaaldin, Rawaz Kurda.

**Supervision:** Rawaz Kurda.

**Writing – original draft:** Bashdar Omer, Mahmood Hunar Dheyaaldin.

**Writing – review & editing:** Najmadeen Mohammed Saeed, Ahmed Salah Jamal, Rawaz Kurda.

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
