## [Decision Letter · Decision Letter 0]

24 Sep 2024

PONE-D-24-20165Evaluating the long-term strength of GGBFS-blended cement across various water-to-binder and superplasticizer ratio under heating/cooling cyclesPLOS ONE

Dear Dr. Kurda,

Thank you for submitting your manuscript to PLOS ONE. After careful consideration, we feel that it has merit but does not fully meet PLOS ONE’s publication criteria as it currently stands. Therefore, we invite you to submit a revised version of the manuscript that addresses the points raised during the review process.

We look forward to receiving your revised manuscript.

Kind regards,

Parthiban Kathirvel

Academic Editor

PLOS ONE

Reviewers' comments:

Reviewer's Responses to Questions

2. We note that your Data Availability Statement is currently as follows: “All relevant data are within the manuscript”

Please confirm at this time whether or not your submission contains all raw data required to replicate the results of your study. Authors must share the “minimal data set” for their submission. PLOS defines the minimal data set to consist of the data required to replicate all study findings reported in the article, as well as related metadata and methods (https://journals.plos.org/plosone/s/data-availability#loc-minimal-data-set-definition ).

If your submission does not contain these data, please either upload them as Supporting Information files or deposit them to a stable, public repository and provide us with the relevant URLs, DOIs, or accession numbers. For a list of recommended repositories, please see https://journals.plos.org/plosone/s/recommended-repositories .

**Comments to the Author**

1. Is the manuscript technically sound, and do the data support the conclusions?

Reviewer #1: Yes

Reviewer #2: Yes

2. Has the statistical analysis been performed appropriately and rigorously? 

Reviewer #1: No

Reviewer #2: Yes

3. Have the authors made all data underlying the findings in their manuscript fully available?

Reviewer #1: No

Reviewer #2: Yes

4. Is the manuscript presented in an intelligible fashion and written in standard English?

Reviewer #1: Yes

Reviewer #2: Yes

5. Review Comments to the Author

Reviewer #1: Abstract

1. What are the method & conclusion?

Experimental Program

1. There are results in Section 2 whereby it should be moved to Section 3.

Result and Discussion

1. Section 3 is wee long & needs to be summarized as the current format is similar to thesis.

Conclusion

1. Section 5 is wee long & needs to be summarized in full sentences without bullet points.

References

1. Please follow the format of the references.

2. There are a number of incomplete particulars such as page number & etc in journals.

Reviewer #2: Abstract:

The first sentence of the abstract can be omitted

Introduction:

[1] The first paragraph of the introduction part is not necessary. The authors can reduce it or can replace with physicochemical characteristics of their parent material GGBFS.

[2] "30 cyclic heating and cooling processes will be used in the study to test

various temperatures in order to incorporate ongoing challenges present in the real world" Suggest some reference to this sentence

[3] Rest part of the introduction is ok

Experimental design

[1] Section 2.1

Mention the manufacturing details (if any) of all the materials used

[2] "Ground Granulated Blast Furnace Slag (GGBFS) with a specific gravity of 2.90, the bulk

density is 1000-1100 kg/m3 and the fineness is 360 more." state the source of this claim

[3] Section 2.5: Authors must write one sentence about the sole purpose of this experimentation. What they wanted to achieve and how would the result help them with their objectives.

[4] Superscripts are missing at some places such as kg/m3, try to correct those in the revised manuscript

Results and discussions:

[1] Section 3.1: "The maximum

flowability was obtained with 0.48 w/b, 2% SP, and 45% GGBFS. It can be shown that higher GGBFS

replacement content led to higher flowability in all mixes" Authors can compare this result with existing literature values

[2] Fig. 12c: Explain why a quadratic regression is necessary, anyways it gives a poor correlation

[3] Fig. 18: EDX/FTIR/XRD data would be more beneficial

Sustainable development:

[1] This section doesn't enhance the objective of the study. The authors can distribute parts of this section into the introduction, results and conclusion.

[2] Sustainable development is a separate study and many stastical parameters are required to be analysed. The authors may think of doing a separate study.

Overall comment:

The authors must compare their results with the existing literature in form of a table. That will help the readers point out their current scenario.

Conclusion:

6. PLOS authors have the option to publish the peer review history of their article (what does this mean? ). If published, this will include your full peer review and any attached files.

**Do you want your identity to be public for this peer review?** For information about this choice, including consent withdrawal, please see our Privacy Policy .

Reviewer #1: No

Reviewer #2: No

---

## [Author Response · Author response to Decision Letter 1]

12 Oct 2024

Reviewer #1: Reviewer comments

1- Abstract: The abstract should be improved and include clear statements about objectives, methodology and findings.

Reply:

The abstract has been altered according to the reviewer’s comment.

2. Relationship to Literature: The paper is bad structured and poorly recent documented, the amount of data is insufficient. It is necessary more recent and appropriate literature. Review more published studies to enhance the background the present study. Recent related references need to be incorporated. Introduction paragraph one needs proper references.

Reply: as the reviewer requested, more recent references have been added to the introduction.

3. Methodology: The methods employed are robust and appropriate but more was needed in order to arrive a more information and results. Rebuild the section (method and experimental (RSM)- results and discussion).

Reply: In order to make the introduction clearer, the first paragraph of section 2 was altered.

4. It is better to provide the images of the specimens prepared for the test.

Reply: please, see Figure 3. The specimens were shown.

5. the paper only describes results and does not discuss findings with other studies.

Reply: The authors agree with the reviewer’s comment and this step is an important parameter to be considered for any research. Nevertheless, after conducting an in-depth literature review, it was revealed that there is little to no data on the evaluation of GGBFS blended cement under different GGBFS replacement contents, w/b ratios, and SP dosages. Thus, we could not make a comparison between the data of this study and the others.

6 . A flowchart should be provided for the work process. The flow chart of the study has to be described in the steps.

Reply: as shown in Figure 3 (Fig 1. A schematic of the experimental process.), In this study, we tried to focus on the work process instead the whole process. Instead of the flowchart, we described the work process of the work as a text (see, the first paragraph of section 2).

7. Why is it based only on the compressive strength (where are the other mechanical strengths)?

Reply: IN this work, we studied 27 different mixes. These specimens were then removed after 24 hours and subjected to water curing for 28, 56, 90, and 365 days. Our objective to mainly focus on the compressive strength. Other tests can be added for future works such as a book.

8. improve the presentability of figures 12a to 12c (badly illustrated)

Reply:

Based on the reviewers’ comment, the figures have been changed and the trend line was added to understand the trend.

Reviewer #2:

The manuscript was crafted in a distinctive way in which the presented experimental results were validated using response surface methodology as well as Anova. Both statistical results reinforced and backed the findings of the author, as expected, that every experimental result should have either statistical, simulation, or machine learning model validation.

The typing error in line 5 under the flowability of result and discussion heading should be corrected from "ADDINg" to "adding."

Reply: Thank you very much for your valuable comment. As the reviewer requested, the word “ADDing” was corrected.

Reviewer #3:

Doesn’t meet the criteria for publication. Infect doesn’t have required elements of a research paper.

A superficial lab test in terms of mixture test (Table 1), use of many undefined acronyms, , inadequate samples, … In terms of innovation doesn’t have any advantage over

https://iopscience.iop.org/article/10.1088/1755-1315/498/1/012045,

https://www.sciencedirect.com/science/article/pii/S2666790823000095

Reply: Both papers have been considered in the study.

Typically disorganized,

1. Long and vague title

Reply: the tittle has been altered according to the

2. Poor English with massive flaws which must extensively be revised by a native expert

Reply: The English has been checked again and we only found a minor

3. Not critically analyzed relevant works. Simply the gap can be seen in reference list.

4. Lack of any Discussion, solid comparison, limitation, practical and pitfall difficulties

8. Obvious inadequate sample for any generalizing, statistical analysis

Reply: comments 4 and 8 in which they are similar. We decided to reply them jointly. The discussion section was 25 pages. This is way more than any usual paper. We don’t mind to discuss more but the limitation of the journal gaudiness may not allow that.

5. Outdated references with almost belong to more than two decades ago

Reply: The list of the references has been updated

6. Unhighlighted research gaps with at least 3 years gaps!!!! WONDERED no work after 2021????? See the given example above.

Reply: The relevant studies have been updated

7. Pretty long and unjustified conclusion.

Reply: Due to the fact that we considered many references, mixes, the conclusion was needed to be long.

9. Many inappropriate citations.

Reply: The list of the references has been updated

10. Ill-formatted and inconsistent reference list

Reply: The list of the references has been updated

---

## [Decision Letter · Decision Letter 1]

19 Nov 2024

PONE-D-24-20165R1Evaluating the long-term strength of GGBFS-blended cement across various water-to-binder and superplasticizer ratio under heating/cooling cyclesPLOS ONE

Dear Dr. Kurda,

Thank you for submitting your manuscript to PLOS ONE. After careful consideration, we feel that it has merit but does not fully meet PLOS ONE’s publication criteria as it currently stands. Therefore, we invite you to submit a revised version of the manuscript that addresses the points raised during the review process.

We look forward to receiving your revised manuscript.

Kind regards,

Parthiban Kathirvel

Academic Editor

PLOS ONE

Journal Requirements:

Reviewers' comments:

Reviewer's Responses to Questions

**Comments to the Author**

1. If the authors have adequately addressed your comments raised in a previous round of review and you feel that this manuscript is now acceptable for publication, you may indicate that here to bypass the “Comments to the Author” section, enter your conflict of interest statement in the “Confidential to Editor” section, and submit your "Accept" recommendation.

Reviewer #3: (No Response)

2. Is the manuscript technically sound, and do the data support the conclusions?

Reviewer #3: Yes

3. Has the statistical analysis been performed appropriately and rigorously? 

Reviewer #3: Yes

4. Have the authors made all data underlying the findings in their manuscript fully available?

Reviewer #3: Yes

5. Is the manuscript presented in an intelligible fashion and written in standard English?

Reviewer #3: No

6. Review Comments to the Author

Reviewer #3: Major Comments

Title:

The title is overly long and lacks clarity. Consider simplifying it while maintaining focus on the main research objectives.

Introduction:

While the introduction highlights the importance of GGBFS, the literature review needs better integration of recent studies. Several references are outdated and lack representation of the latest advancements post-2021.

The research gap is vaguely defined. Strengthen the justification for the study by identifying more precise gaps in existing literature.

Figures and Tables:

Figures 12a-12c and others lack clarity and are poorly illustrated. Improve their resolution, consistency in formatting, and ensure they convey the intended data trends clearly.

Tables contain significant data but need clearer captions. Ensure alignment with journal standards for table formatting.

Language:

The manuscript contains several grammatical errors and typographical issues. A professional language editing service is highly recommended to improve readability.

Conclusion:

The conclusions section is overly lengthy and repetitive. Focus on key takeaways and implications of the findings for the construction industry and sustainability.

Minor Comments

Some acronyms, such as SP and w/b, are not defined consistently throughout the manuscript. Ensure these are explained upon first usage.

Highlight the practical implications of using GGBFS in different environmental and structural settings.

The limitations of the study are not discussed adequately. Address sample size, generalizability, and experimental constraints.

7. PLOS authors have the option to publish the peer review history of their article (what does this mean? ). If published, this will include your full peer review and any attached files.

**Do you want your identity to be public for this peer review?** For information about this choice, including consent withdrawal, please see our Privacy Policy .

Reviewer #3: No

---

## [Author Response · Author response to Decision Letter 2]

6 Feb 2025

Reviewer #3: Major Comments

Comment #1

Title:

The title is overly long and lacks clarity. Consider simplifying it while maintaining focus on the main research objectives.

Response: Thanks for your valuable comments. Kindly the title of research modified and reduced the length.

Comment #2

Introduction:

While the introduction highlights the importance of GGBFS, the literature review needs better integration of recent studies. Several references are outdated and lack representation of the latest advancements post-2021.

Response: Thanks for your valuable comments. Numerous research studies regarding GGBSF replacement to cement performance have been conducted and explained in detail with the latest research studies.

Comment #3

The research gap is vaguely defined. Strengthen the justification for the study by identifying more precise gaps in existing literature.

Response: Thanks for your valuable comments, The research gaps have been kindly explained. The future perspective has been added and combined with the Research gap paragraph.

Comment #4

Figures and Tables:

Figures 12a-12c and others lack clarity and are poorly illustrated. Improve their resolution, consistency in formatting, and ensure they convey the intended data trends clearly.

Tables contain significant data but need clearer captions. Ensure alignment with journal standards for table formatting.

Response: Thanks for your valuable comments. Kindly, Figure 12 has been entirely modified and the resolution of the figures has improved

Comment #5

Language:

The manuscript contains several grammatical errors and typographical issues. A professional language editing service is highly recommended to improve readability.

Response: Thanks for your valuable comments. All the grammatical errors corrected and entire manuscript checked for any typo issues and structure of sentences have been improved.

Comment #6

Conclusion:

The conclusions section is overly lengthy and repetitive. Focus on key takeaways and implications of the findings for the construction industry and sustainability.

Response: Thanks for your valuable comments, the conclusion section has been entirely improved. And the key points of finding have been explained.

Comment #7

Some acronyms, such as SP and w/b, are not defined consistently throughout the manuscript. Ensure these are explained upon first usage.

Highlight the practical implications of using GGBFS in different environmental and structural settings.

The limitations of the study are not discussed adequately. Address sample size, generalizability, and experimental constraints.

Response: Thanks for your valuable comments, The acronyms have been explained for the first use of explanation from abstract and Introduction. And regarding GGBFS a numerous of explanation of GGBFS affect included performance on different environments have kindly explained.

---

## [Editor Report · Decision Letter 2]

11 Feb 2025

Evaluating the long-term strength of GGBFS-blended cement across various water-to-binder and superplasticizer ratios under heating/cooling cycles

PONE-D-24-20165R2

Dear Dr. Kurda,

We’re pleased to inform you that your manuscript has been judged scientifically suitable for publication and will be formally accepted for publication once it meets all outstanding technical requirements.

Kind regards,

Parthiban Kathirvel

Academic Editor

PLOS ONE

---

## [Editor Report · Acceptance letter]

PONE-D-24-20165R2

PLOS ONE

Dear Dr. Kurda,

I'm pleased to inform you that your manuscript has been deemed suitable for publication in PLOS ONE. Congratulations! Your manuscript is now being handed over to our production team.

Kind regards,

on behalf of

Dr. Parthiban Kathirvel

Academic Editor

PLOS ONE